# StyleNeRF: A Style-based 3D-Aware Generator for High-resolution Image Synthesis

**Jiatao Gu[†], Lingjie Liu[‡*], Peng Wang[◇], Christian Theobalt[‡]**

[†]Meta AI   [‡]Max Planck Institute for Informatics   [◇]The University of Hong Kong

[†]jgu@fb.com   [‡]{lliu,theobalt}@mpi-inf.mpg.de   [◇]pwang3@cs.hku.hk

## Abstract

We propose *StyleNeRF*, a 3D-aware generative model for photo-realistic high-resolution image synthesis with high multi-view consistency, which can be trained on unstructured 2D images. Existing approaches either cannot synthesize high-resolution images with fine details or yield noticeable 3D-inconsistent artifacts. In addition, many of them lack control over style attributes and explicit 3D camera poses. *StyleNeRF* integrates the neural radiance field (NeRF) into a style-based generator to tackle the aforementioned challenges, i.e., improving rendering efficiency and 3D consistency for high-resolution image generation. We perform volume rendering only to produce a low-resolution feature map and progressively apply upsampling in 2D to address the first issue. To mitigate the inconsistencies caused by 2D upsampling, we propose multiple designs, including a better upsampler and a new regularization loss. With these designs, *StyleNeRF* can synthesize high-resolution images at interactive rates while preserving 3D consistency at high quality. *StyleNeRF* also enables control of camera poses and different levels of styles, which can generalize to unseen views. It also supports challenging tasks, including style mixing and semantic editing. Code and pre-trained models are available at: https://github.com/facebookresearch/StyleNeRF.

## 1 Introduction

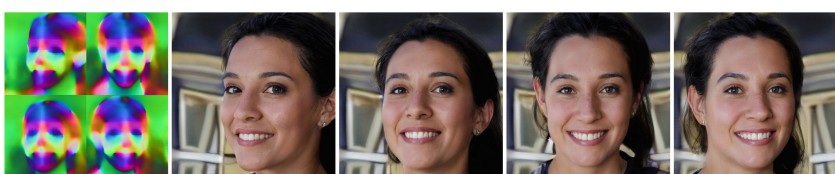

Figure 1: Synthesized $1024^2$ images by *StyleNeRF*, with the low-resolution feature maps.

Photo-realistic free-view image synthesis is a long-standing problem in computer vision and computer graphics. Traditional graphics pipeline requires production-quality 3D models, computationally expensive rendering and manual work, making it challenging to apply to large-scale real-world scenes. In the meantime, Generative Adversarial Networks (GANs, Goodfellow et al., 2014) can be trained on a large number of unstructured images to generate high-quality images. However, most GAN models operate in 2D space and lack the 3D understanding of the input, which results in their inability to synthesize images of the same scene with 3D consistency and direct camera control.

Natural images are the 2D projection of the 3D world. Hence, recent works on generative models (Schwarz et al., 2020; Chan et al., 2021) enforce 3D structures by incorporating a neural radiance field (NeRF, Mildenhall et al., 2020). However, these methods cannot synthesize high-resolution images with delicate details due to the computationally expensive rendering process of NeRF. Furthermore, the slow rendering process leads to inefficient training and makes these models unsuitable for interactive applications. GIRAFFE (Niemeyer & Geiger, 2021b) combines NeRF with a CNN-

---

[*]corresponding author.

based renderer, which has the potential to synthesize high-resolution images. However, this method falls short of 3D-consistent image generation and so far has not shown high-resolution results.

We propose *StyleNeRF*, a new 3D-aware generative model for high-resolution 3D consistent image synthesis at interactive rates. It also allows control of the 3D camera pose and enables control of specific style attributes. *StyleNeRF* incorporates 3D scene representations into a style-based generative model. To prevent the expensive direct color image rendering from the original NeRF approach, we only use NeRF to produce a low-resolution feature map and upsample it progressively to high resolution. To improve 3D consistency, we propose several designs, including a desirable upsampler that achieves high consistency while mitigating artifacts in the outputs, a novel regularization term that forces the output to match the rendering result of the original NeRF and fixing the issues of view direction condition and noise injection. *StyleNeRF* is trained using unstructured real-world images. A progressive training strategy significantly improves the stability of learning real geometry.

We evaluate *StyleNeRF* on various challenging datasets. *StyleNeRF* can synthesize photo-realistic $1024^2$ images at interactive rates while achieving high multi-view consistency. None of the existing methods can achieve both characteristics. Additionally, *StyleNeRF* enables direct control on styles.

## 2 RELATED WORK

**Neural Implicit Fields**    Representing 3D scenes as neural implicit fields has increasingly gained much attention. Michalkiewicz et al. (2019); Mescheder et al. (2019); Park et al. (2019); Peng et al. (2020) predict neural implicit fields with 3D supervision. Some of them (Sitzmann et al., 2019; Niemeyer et al., 2019) assume that the ray color only lies on the geometry surface and propose differentiable renderers to learn a neural implicit surface representation. NeRF and and similar works (Lombardi et al., 2019; Mildenhall et al., 2020; Liu et al., 2020; Zhang et al., 2020) utilize a volume rendering technique to render neural implicit volume representations for novel view synthesis. In this work, we focus on generative NeRF. Unlike the discussed methods, which require posed multi-view images, our approach only needs unstructured single-view images for training.

**Image Synthesis with GANs**    Starting from Goodfellow et al. (2014), GANs have demonstrated high-quality results (Durugkar et al., 2017; Mordido et al., 2018; Doan et al., 2019; Zhang et al., 2019; Brock et al., 2018; Karras et al., 2018). StyleGANs (Karras et al., 2019; 2020b) achieve SOTA quality and support different levels of style control. Karras et al. (2021) solve the "texture sticking" problem of 2D GANs in generating animations with 2D transformations. Some methods (Härkönen et al., 2020; Tewari et al., 2020a; Shen et al., 2020; Abdal et al., 2020; Tewari et al., 2020b; Leimkühler & Drettakis, 2021; Shoshan et al., 2021) leverage disentangled properties in the latent space to enable explicit controls, most of which focus on faces. While these methods can synthesize face poses parameterized by two angles, extending them to general objects and controlling 3D cameras is not easy. Chen et al. (2021a) proposed to generate segmentation maps from implicit fields to enable 3D control. However, it requires 3D meshes for pre-training. In contrast, our work can synthesize images for *general objects*, enabling *explicit 3D camera* control.

**3D-Aware GANs**    Recently, 3D representations have been integrated into 2D generative models to enable camera control. Voxel-based GANs (Henzler et al., 2019; Nguyen-Phuoc et al., 2019; 2020) lack fine details in the output due to resolution restriction. Radiance fields-based methods (Schwarz et al., 2020; Chan et al., 2021; Niemeyer & Geiger, 2021a) achieve higher quality but have difficulties in training on high-resolution images ($512^2$ and beyond) due to the expensive rendering process. GIRAFFE (Niemeyer & Geiger, 2021b) improves the training and rendering efficiency by combining NeRF with a CNN-based renderer; GSN (DeVries et al., 2021) models a locally conditional NeRF with a similar renderer for unconstrained indoor scene generation. However, they both produce severe view-inconsistent artifacts due to their network designs (e.g., $3 \times 3$ `Conv` and upsampler). In contrast, our method can effectively preserve view consistency in image synthesis.

## 3 METHOD

### 3.1 IMAGE SYNTHESIS AS NEURAL IMPLICIT FIELD RENDERING

**Style-based Generative Neural Radiance Field**    We start by modeling a 3D scene as neural radiance field (NeRF, Mildenhall et al., 2020). It is typically parameterized as multilayer perceptrons

(MLPs), which takes the position $\boldsymbol{x} \in \mathbb{R}^3$ and viewing direction $\boldsymbol{d} \in \mathbb{S}^2$ as input, and predicts the density $\sigma(\boldsymbol{x}) \in \mathbb{R}^+$ and view-dependent color $\boldsymbol{c}(\boldsymbol{x}, \boldsymbol{d}) \in \mathbb{R}^3$. To model high-frequency details, follwing NeRF (Mildenhall et al., 2020), we map each dimension of $\boldsymbol{x}$ and $\boldsymbol{d}$ with Fourier features :

$$\zeta^L(x) = \left[ \sin(2^0 x), \cos(2^0 x), \dots, \sin(2^{L-1} x), \cos(2^{L-1} x) \right] \tag{1}$$

We formalize *StyleNeRF* representations by conditioning NeRF with style vectors $\boldsymbol{w}$ as follows:

$$\phi_{\boldsymbol{w}}^n(\boldsymbol{x}) = g_{\boldsymbol{w}}^n \circ g_{\boldsymbol{w}}^{n-1} \circ \dots \circ g_{\boldsymbol{w}}^1 \circ \zeta(\boldsymbol{x}), \text{ where } \boldsymbol{w} = f(\boldsymbol{z}), \boldsymbol{z} \in \mathcal{Z} \tag{2}$$

Similar as StyleGAN2 (Karras et al., 2020b), $f$ is a mapping network that maps noise vectors from the spherical Gaussian space $\mathcal{Z}$ to the style space $\mathcal{W}$; $g_{\boldsymbol{w}}^i(.)$ is the $i^{\text{th}}$ layer MLP whose weight matrix is modulated by the input style vector $\boldsymbol{w}$. $\phi_{\boldsymbol{w}}^n(\boldsymbol{x})$ is the $n$-th layer feature of that point. We then use the extracted features to predict the density and color, respectively:

$$\sigma_{\boldsymbol{w}}(\boldsymbol{x}) = h_\sigma \circ \phi_{\boldsymbol{w}}^{n_\sigma}(\boldsymbol{x}), \quad \boldsymbol{c}_{\boldsymbol{w}}(\boldsymbol{x}, \boldsymbol{d}) = h_c \circ [\phi_{\boldsymbol{w}}^{n_c}(\boldsymbol{x}), \zeta(\boldsymbol{d})], \tag{3}$$

where $h_\sigma$ and $h_c$ can be a linear projection or 2-layer MLPs. Different from the original NeRF, we assume $n_c > n_\sigma$ for Equation (3) as the visual appearance generally needs more capacity to model than the geometry. The first $\min(n_\sigma, n_c)$ layers are shared in the network.

**Volume Rendering**  Image synthesis is modeled as volume rendering from a given camera pose $\boldsymbol{p} \in \mathcal{P}$. For simplicity, we assume a camera is located on the unit sphere pointing to the origin with a fixed field of view (FOV). We sample the camera's pitch & yaw from a uniform or Gaussian distribution. To render an image $I \in \mathbb{R}^{H \times W \times 3}$, we shoot a camera ray $\boldsymbol{r}(t) = \boldsymbol{o} + t\boldsymbol{d}$ ($\boldsymbol{o}$ is the camera origin) for each pixel, and then calculate the color using the volume rendering equation:

$$I_{\boldsymbol{w}}^{\text{NeRF}}(\boldsymbol{r}) = \int_0^\infty p_{\boldsymbol{w}}(t) \boldsymbol{c}_{\boldsymbol{w}}(\boldsymbol{r}(t), \boldsymbol{d}) dt, \text{ where } p_{\boldsymbol{w}}(t) = \exp\left( -\int_0^t \sigma_{\boldsymbol{w}}(\boldsymbol{r}(s)) ds \right) \cdot \sigma_{\boldsymbol{w}}(\boldsymbol{r}(t)) \tag{4}$$

In practice, the above equation is discretized by accumulating sampled points along the ray. Following NeRF (Mildenhall et al., 2020), stratified and hierarchical sampling are applied for more accurate discrete approximation to the continuous implicit function.

**Challenges**  Compared to 2D generative models (e.g., StyleGANs (Karras et al., 2019; 2020b)), the images generated by NeRF-based models have 3D consistency, which is guaranteed by modeling the image synthesis as a physics process, and the neural 3D scene representation is invariant across different viewpoints. However, the drawbacks are apparent: these models cost much more computation to render an image at the exact resolution. For example, 2D GANs are $100 \sim 1000$ times more efficient to generate a $1024^2$ image than NeRF-based models. Furthermore, NeRF consumes much more memory to cache the intermediate results for gradient back-propagation during training, making it difficult to train on high-resolution images. Both of these restrict the scope of applying NeRF-based models in high-quality image synthesis, especially at the training stage when calculating the objective function over the whole image is crucial.

## 3.2 Approximation for high-resolution image generation

In this section, we propose how to improve the efficiency of *StyleNeRF* by taking inspiration from 2D GANs. We observe that the image generation of 2D GANs (e.g., StyleGANs) is fast due to two main reasons: (1) each pixel only takes single forward pass through the network; (2) image features are generated progressively from coarse to fine, and the feature maps with higher resolutions typically have a smaller number of channels to save memory.

In *StyleNeRF*, the first point can be partially achieved by early aggregating the features into the 2D space before the final colors are computed. In this way, each pixel is assigned with a feature vector, Furthermore, it only needs to pass through a network once rather than calling the network multiple times for all sampled points on the ray as NeRF does. We approximate Equation (4) as:

$$I_{\boldsymbol{w}}^{\text{Approx}}(\boldsymbol{r}) = \int_0^\infty p_{\boldsymbol{w}}(t) \cdot h_c \circ [\phi_{\boldsymbol{w}}^{n_c}(\boldsymbol{r}(t)), \zeta(\boldsymbol{d})] \, dt \approx h_c \circ [\phi_{\boldsymbol{w}}^{n_c, n_\sigma}(\mathcal{A}(\boldsymbol{r})), \zeta(\boldsymbol{d})], \tag{5}$$

where $\phi_{\boldsymbol{w}}^{n, n_\sigma}(\mathcal{A}(\boldsymbol{r})) = g_{\boldsymbol{w}}^n \circ g_{\boldsymbol{w}}^{n-1} \circ \dots \circ g_{\boldsymbol{w}}^{n_\sigma+1} \circ \mathcal{A}(\boldsymbol{r})$ and $\mathcal{A}(\boldsymbol{r}) = \int_0^\infty p_{\boldsymbol{w}}(t) \cdot \phi_{\boldsymbol{w}}^{n_\sigma}(\boldsymbol{r}(t)) dt$. The definitions of $\mathcal{A}(.)$ and $\phi_{\boldsymbol{w}}^{n, n_\sigma}(.)$ can be extended to the operations on a set of rays, each ray

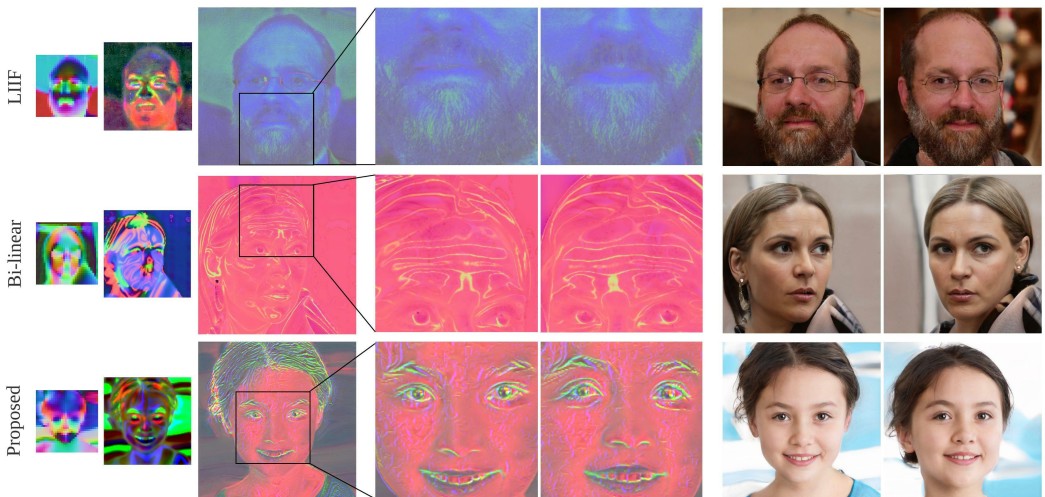

Figure 2: Internal representations and the outputs from *StyleNeRF* trained with different upsampling operators. With LIIF, patterns stick to the same pixel coordinates when the viewpoint changes; With bilinear interpolation, bubble-shape artifacts can be seen on the feature maps and images. Our proposed upsampler faithfully preserves 3D consistency while getting rid of bubble-shape artifacts.

processed independently. Next, instead of using volume rendering to render a high-resolution feature map directly, we can employ NeRF to generate a downsampled feature map at a low resolution and then employ upsampling in 2D space to progressively increase into the required high resolution. We take two adjacent resolutions as an example. Suppose $R_L \in \mathbb{R}^{H/2 \times W/2}$ and $R_H \in \mathbb{R}^{H \times W}$ are the corresponding rays of the pixels in the low- and high-resolution images, respectively. To approximate the high-resolution feature map, we can up-sample in the low-resolution feature space:

$$\phi_{\boldsymbol{w}}^{n,n_\sigma}(\mathcal{A}(R_H)) \approx \texttt{Upsample}\left(\phi_{\boldsymbol{w}}^{n,n_\sigma}(\mathcal{A}(R_L))\right) \qquad (6)$$

Recursively inserting `Upsample` operators enables efficient high-resolution image synthesis as the computationally expensive volume rendering only needs to generate a low-resolution feature map. The efficiency is further improved when using fewer channels for higher resolution.

While early aggregation and upsampling operations can accelerate the rendering process for high-resolution image synthesis, they would come with scarification to the inherent consistency of NeRF. There are two reasons why they introduce inconsistency. First, the resulting model contains non-linear transformations to capture spurious correlations in 2D observation, mainly when substantial ambiguity exists. For example, our training data are unstructured single-view images without sufficient multi-view supervision. Second, such a pixel-space operation like up-sampling would compromise 3D consistency. Therefore, naïve model designs would lead to severe multi-view inconsistent outputs (e.g., when moving the camera to render images, hairs are constantly changing). In the following, we propose several designs and choices to alleviate the inconsistency in the outputs.

### 3.3 PRESERVING 3D CONSISTENCY

**Upsampler design** Up-sampling in 2D space causes multi-view inconsistency in general; however, the specific design choice of the upsampler determines how much such inconsistency is introduced. As our model is directly derived from NeRF, MLP ($1 \times 1$ `Conv`) is the basic building block. With MLPs, however, pixel-wise learnable upsamplers such as pixelshuffle (Shi et al., 2016) or LIIF (Chen et al., 2021b) produce "chessboard" or "texture sticking" artifacts due to its tendency of relying on the image coordinates implicitly. In the meanwhile, Karras et al. (2019; 2020b; 2021) proposed to use non-learnable upsamplers that interpolate the feature map with pre-defined low-pass filters (e.g. bilinear interpolation). While these upsamplers can produce smoother outputs, we observed non-removable "bubble" artifacts in both the feature maps and output images. We conjecture it is due to the lack of local variations when combining MLPs with fixed low-pass filters. We achieve the balance between consistency and image quality by combining these two approaches (see Figure 2). For any input feature map $X \in \mathbb{R}^{N \times N \times D}$:

$$\texttt{Upsample}(X) = \texttt{Conv2d}\left(\texttt{Pixelshuffle}\left(\texttt{Repeat}(X,4) + \psi_\theta(X), 2\right), K\right), \qquad (7)$$

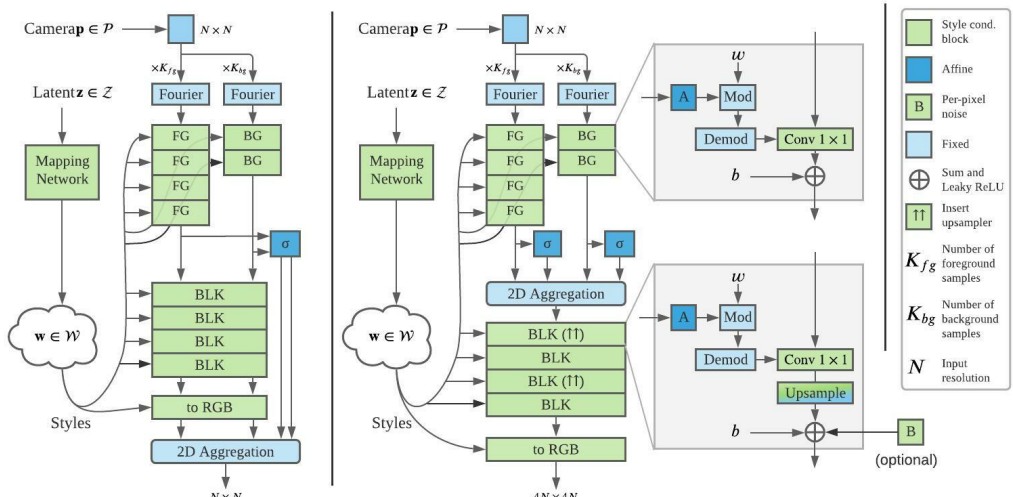

Figure 3: Example architecture. (Left) Original NeRF path; (right) Main path of StyleNeRF.

where $\psi_\theta : \mathbb{R}^D \to \mathbb{R}^{4D}$ is a learnable 2-layer MLP, and $K$ is a fixed blur kernel (Zhang, 2019).

**NeRF path regularization**   We propose a new regularization term to enforce 3D consistency, which regularizes the model output to match the original path (Equation (4)). In this way, the final outputs can be closer to the NeRF results, which have multi-view consistency. This is implemented by sub-sampling pixels on the output and comparing them against those generated by NeRF:

$$\mathcal{L}_{\text{NeRF-path}} = \frac{1}{|S|} \sum_{(i,j) \in S} \left( I_{\boldsymbol{w}}^{\text{Approx}}(R_{\text{in}})[i,j] - I_{\boldsymbol{w}}^{\text{NeRF}}(R_{\text{out}}[i,j]) \right)^2, \tag{8}$$

where $S$ is the set of randomly sampled pixels; $R_{\text{in}}$ and $R_{\text{out}}$ are the corresponding rays of the pixels in the low-resolution image generated via NeRF and high-resolution output of StyleNeRF.

**Remove view direction condition**   Predicting colors with view direction condition was suggested by the original NeRF for modeling view-dependent effects and was by default applied in most follow-up works (Chan et al., 2021; Niemeyer & Geiger, 2021b). However, this design would give the model additional freedom to capture spurious correlations and dataset bias, especially if only a single-view target is provided. We, therefore, remove the view direction in color prediction, which improves the synthesis consistency (See Figure 8).

**Fix 2D noise injection**   Existing studies (Karras et al., 2019; 2020b; Feng et al., 2020) have showed that injecting per-pixel noise can increase the model's capability of modeling stochastic variation (e.g. hairs, stubble). Nevertheless, such 2D noise only exists in the image plane, which will not change in a 3D consistent way when the camera moves. To preserve 3D consistency, our default solution is to trade the model's capability of capturing variation by removing the noise injection as Karras et al. (2021) does. Optionally, we also propose a novel geometry-aware noise injection based on the estimated surface from StyleNeRF. See Appendix A.4 for more details.

### 3.4   STYLENERF ARCHITECTURE

In this section, we describe the network architecture and the learning procedure of *StyleNeRF*.

**Mapping Network**   Following StyleGAN2, latent codes are sampled from standard Gaussian and processed by a mapping network. Finally, the output vectors are broadcast to the synthesis networks.

**Synthesis Network**   Considering that our training images generally have unbounded background, we choose NeRF++ (Zhang et al., 2020), a variant of NeRF, as the *StyleNeRF* backbone. NeRF++ consists of a foreground NeRF in a unit sphere and a background NeRF represented with inverted

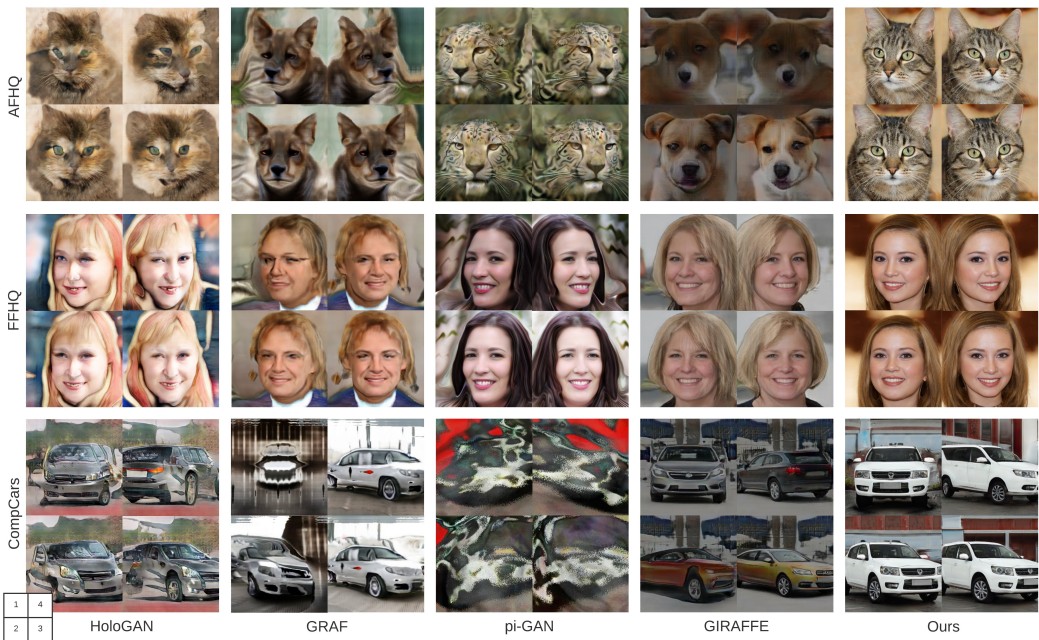

Figure 4: Qualitative comparisons at $256^2$. *StyleNeRF* achieves the best quality and 3D consistency.

sphere parameterization. As shown in Figure 3, two MLPs are used to predict the density where the background network has fewer parameters than the foreground one. Then a shared MLP is employed for color prediction. Each style-conditioned block consists of an affine transformation layer and a $1 \times 1$ convolution layer (`Conv`). The `Conv` weights are modulated with the affine-transformed styles, and then demodulated for computation. `leaky_ReLU` is used as non-linear activation. The number of blocks depends on the input and target image resolutions.

**Discriminator & Objectives**   We use the same discriminator as StyleGAN2. Following previous works (Chan et al., 2021; Niemeyer & Geiger, 2021b), *StyleNeRF* adopts a non-saturating GAN objective with R1 regularization (Mescheder et al., 2018). A new NeRF path regularization is employed to enforce 3D consistency. The final loss function is defined as follows ($D$ is the discriminator and $G$ is the generator including the mapping and synthesis networks):

$$\mathcal{L}(D, G) = \mathbb{E}_{\boldsymbol{z} \sim \mathcal{Z}, \boldsymbol{p} \sim \mathcal{P}} \left[ f(D(G(\boldsymbol{z}, \boldsymbol{p}))) \right] + \mathbb{E}_{I \sim p_{\text{data}}} \left[ f(-D(I) + \lambda \|\nabla D(I)\|^2) \right] + \beta \cdot \mathcal{L}_{\text{NeRF-path}} \quad (9)$$

where $f(u) = -\log(1 + \exp(-u))$, and $p_{\text{data}}$ is the data distribution. We set $\beta = 0.2$ and $\lambda = 0.5$.

**Progressive Training**   We train *StyleNeRF* progressively from low to high resolution, which makes the training more stable and efficient. We observed in the experiments that were directly training for the highest resolution easily makes the model fail to capture the object geometry. We suspect it is because both Equations (5) and (6) are just approximations to the original NeRF. Therefore, inspired by Karras et al. (2017), we propose a new *three*-stage progressive training strategy: For the first $T_1$ images, we train *StyleNeRF* without approximation at low-resolution; then, during $T_1 \sim T_2$ images, both the generator and discriminator linearly increase the output resolutions until reaching the target resolution; At last, we fix the architecture and continue training the model at the highest resolution until $T_3$ images. Please refer to the Appendix A.5 for more details.

## 4 EXPERIMENTS

### 4.1 EXPERIMENTAL SETTINGS

**Datasets**   We evaluate *StyleNeRF* on four high-resolution unstructured real datasets: FFHQ (Karras et al., 2019), MetFaces (Karras et al., 2020a), AFHQ (Choi et al., 2020) and CompCars (Yang et al., 2015). The dataset details are described in Appendix B.

**Baselines** We compare our approach with a voxel-based method, HoloGAN (Nguyen-Phuoc et al., 2019), and three radiance field-based methods: GRAF (Schwarz et al., 2020), $\pi$-GAN (Chan et al., 2021) and GIRAFFE (Niemeyer & Geiger, 2021b). As most of the baselines are restricted to low resolutions, we made the comparison at $256^2$ pixels for fairness. We also report the results of the state-of-the-art 2D GAN (Karras et al., 2020b) for reference. See more details in Appendix C.

**Configurations** All datasets except MetFaces are trained progressively. However, the MetFaces dataset is too small to train stably, so we finetune from the pretrained model for FFHQ at the highest resolution. By default, we train 64 images per batch, and set $T_1 = 500k, T_2 = 5000k$ and $T_3 = 25000k$ images, respectively. The input resolution is fixed $32^2$ for all experiments.

## 4.2 RESULTS

**Qualitative comparison** We evaluate our approach and the baselines on three datasets: FFHQ, AFHQ, and CompCars. Target images are resized to $256^2$. We render each object in a sequence and sample four viewpoints shown counterclockwise in Figure 4. While all baselines can generate images under direct camera control, HoloGAN, GARF, and $\pi$-GAN fail to learn geometry correctly and thus produce severe artifacts. GIRAFFE synthesizes images in better quality; however, it produces 3D inconsistent artifacts: the shape and appearance in the output change constantly when the camera moves. We believe it is due to the wrong choice of $3 \times 3$ `Conv` layers. Compared to the baselines, *StyleNeRF* achieves the best visual quality with high 3D consistency across views. Please see more results in the supplemental video.

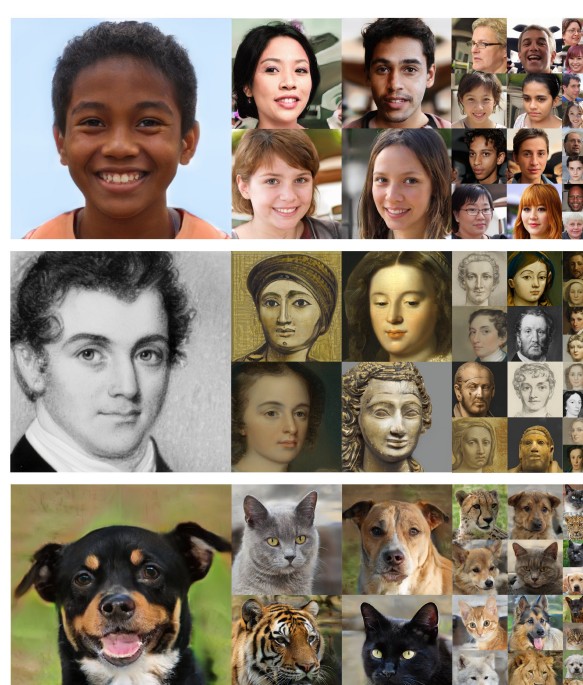

Figure 5: Uncurated set of images at $512^2$ produced by *StyleNeRF* from three datasets. Each example is rendered from a randomly sampled camera.

**Quantitative comparison** We measure the visual quality of image generation by the Frechet Inception Distance (FID, Heusel et al., 2017) and Kernal Inception Distance (KID, Bińkowski et al., 2018) in Table 1. Across all three datasets, *StyleNeRF* consistently outperforms the baselines by significant gains in terms of FID and KID and largely reduces the performance gap between the 3D-aware GANs and the SOTA 2D GAN (i.e., StyleGAN2). Note that while 2D GANs can achieve high quality for each image, they cannot synthesize images of the same scene with 3D consistency. Table 1 also shows the speed comparison over different image resolutions on FFHQ. *StyleNeRF* enables rendering at interactive rates and achieves significant speed-up over voxel and pure NeRF-based methods, and is comparable to GIRAFFE and StyleGAN2. In addition, we include comparison on the consistency of synthesis quantitatively in Appendix D.

**High-resolution synthesis** Unlike the baseline models, our method can generate high-resolution images ($512^2$ and beyond). Figure 5 shows an uncurated set of images rendered by *StyleNeRF*. We also show more results of *StyleNeRF* and report the quantitative results of the SOTA 2D GAN (StyleGAN2 (Karras et al., 2020b)) for reference in the Appendix D.

## 4.3 CONTROLLABLE IMAGE SYNTHESIS

**Explicit camera control** Our method can synthesize novel views with direct camera control and generalize to extreme camera poses, which starkly differs from the training camera pose distribution. Figure 6 shows our results with extreme camera poses, such as zoom-in and -out, and steep view angles. The rendered images maintain good consistency given different camera poses.

Table 1: Quantitative comparisons at $256^2$. We calculate FID, KID$\times 10^3$, and the average rendering time (batch size = 1). The 2D GAN (StyleGAN2 (Karras et al., 2020b)) results are for reference.

| Models | FFHQ $256^2$ | | AFHQ $256^2$ | | CompCars $256^2$ | | Rendering time (ms / image) | | | | |
|---|---|---|---|---|---|---|---|---|---|---|---|
| | FID | KID | FID | KID | FID | KID | 64 | 128 | 256 | 512 | 1024 |
| 2D GAN | 4 | 1.1 | 9 | 2.3 | 3 | 1.6 | - | - | 46 | 51 | 53 |
| HoloGAN | 75 | 68.0 | 78 | 59.4 | 48 | 39.6 | 213 | 215 | 222 | - | - |
| GRAF | 71 | 57.2 | 121 | 83.8 | 101 | 86.7 | 61 | 246 | 990 | 3852 | 15475 |
| $\pi$-GAN | 85 | 90.0 | 47 | 29.3 | 295 | 328.9 | 58 | 198 | 766 | 3063 | 12310 |
| GIRAFFE | 35 | 23.7 | 31 | 13.9 | 32 | 23.8 | 8 | - | 9 | - | - |
| Ours | 8 | 3.7 | 14 | 3.5 | 8 | 4.3 | - | - | 65 | 74 | 98 |

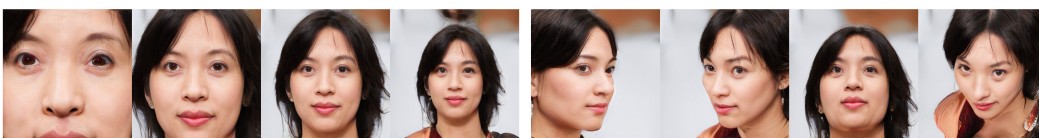

Figure 6: Images synthesized from camera poses which starkly differ from training camera poses.

**Style mixing and interpolation**  Figure 7 shows the results of style mixing and interpolation. As shown in style mixing experiments, copying styles before 2D aggregation affects geometry aspects (shape of noses, glasses, etc.), while copying those after 2D aggregation brings changes in appearance (colors of skins, eyes, hairs, etc.), which indicates clear disentangled styles of geometry and appearance. In the style interpolation results, the smooth interpolation between two different styles without visual artifacts further demonstrates that the style space is semantically learned.

**Style inversion and editing**  We can also do inverse rendering tasks with a learned *StyleNeRF* model. We first pre-train a camera pose predictor in a self-supervised manner. Then given an input image, we use this predictor to estimate the camera pose and optimize the styles via back-propagation to find the best styles that match the target image. We can further perform semantic editing. For instance, we can optimize the styles according to text snippet using a CLIP loss (Radford et al., 2021; Patashnik et al., 2021). An example is shown in Figure 7.

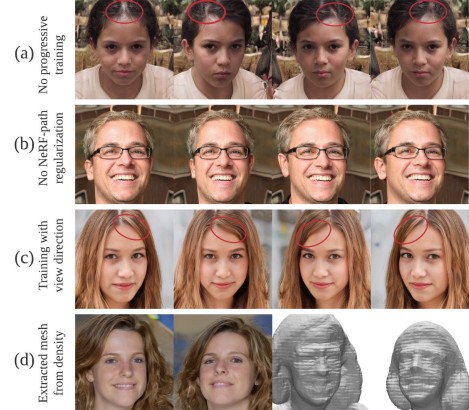

Figure 8: Failure results.

### 4.4 ABLATION STUDIES

Figure 8 (a) shows a typical result without progressive training. Despite not directly affecting the quality of each single image, the model fails to learn the correct shape, e.g. the face should be in convex shape but is predicted as concave surfaces. This leads to severe 3D inconsistent artifacts when the camera moves (see the red circles in Figure 8 (a))

As shown in Figure 8 (b), without NeRF-path regularization, the model sometimes gets stuck learning geometry in 3D and produces a "flat shape" output.

Figure 8 (c) demonstrates an example of the result with view direction as a condition. The modeling of view-dependent effects introduces an ambiguity between 3D shape and radiance and thus leads to a degenerate solution without multi-view consistency (see the red circles in Figure 8 (c)).

### 4.5 LIMITATIONS AND FUTURE WORK

While *StyleNeRF* can efficiently synthesize photo-realistic high-resolution images in high multi-view consistency with explicit camera control, it sacrifices some properties that pure NeRF-based

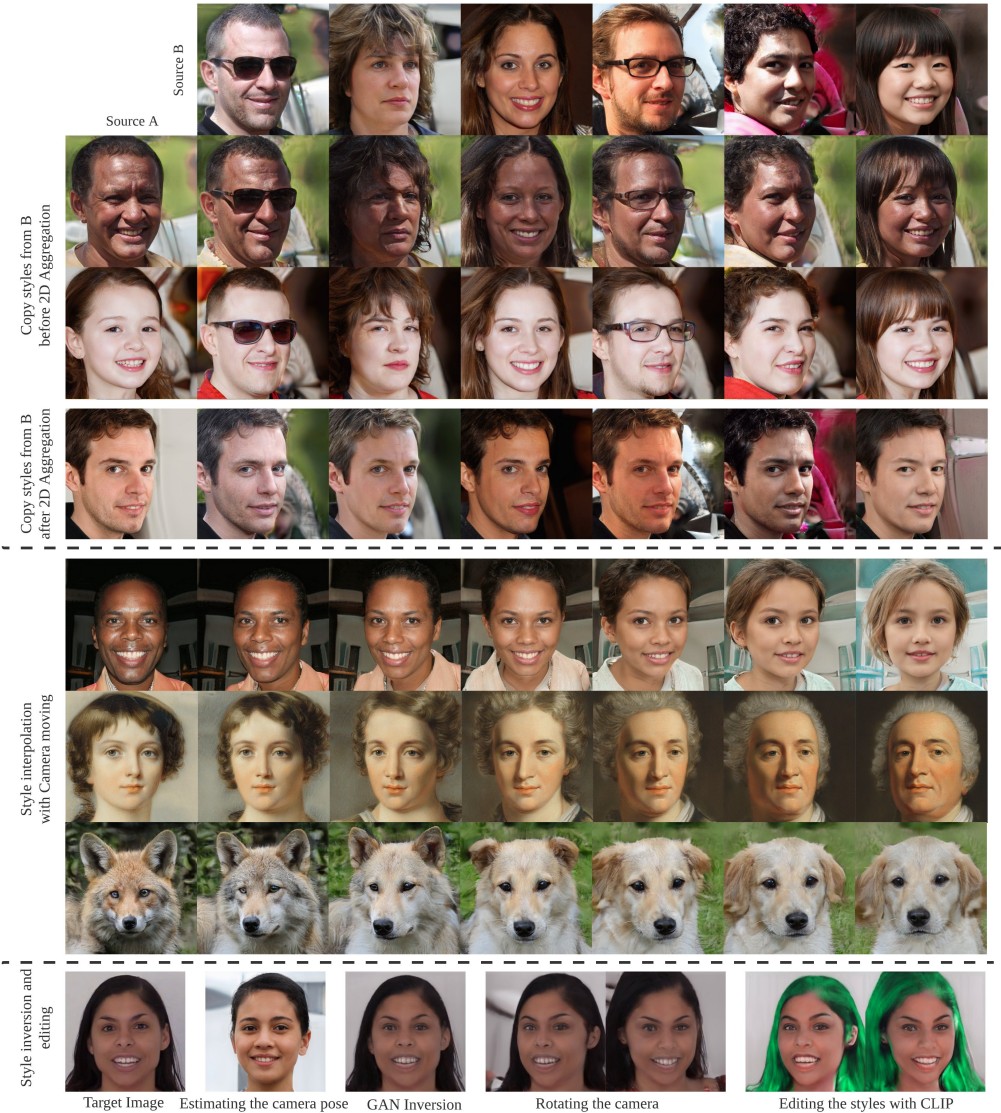

Figure 7: *Style mixing (Top)*: images were generated by copying the specified styles from source B to source A. All images are rendered from the same camera pose. *Style interpolation (Middle)*: we linearly interpolate two sets of style vectors (leftmost and rightmost images) while rotating the camera. *Style inversion and editing (Below)*: the target image is selected from the DFDC dataset (Dolhansky et al., 2019). To edit with CLIP scores, we input "a person with green hair" as the target text.

methods have. An example is shown in Figure 8 (d), where we extract the underlying geometry with marching cube from the learned density. Although *StyleNeRF* is able to recover a coarse geometry, it captures less details compared to the pure NeRF-based models such as $\pi$-GAN (Chan et al. (2021)). Additionally, *StyleNeRF* only empirically preserves multi-view consistency, but it does not guarantee strict 3D consistency. Further exploration and theoretical analysis are needed.

## 5    CONCLUSION

We proposed a 3D-aware generative model, *StyleNeRF*, for efficient high-resolution image generation with high 3D consistency, which allows control over explicit 3D camera poses and style attributes. Our experiments have demonstrated that *StyleNeRF* can synthesize photo-realistic $1024^2$ images at interactive rates and outperforms previous 3D-aware generative methods.

## ETHICS STATEMENT

Our work focuses on technical development, i.e., synthesizing high-quality images with user control. Our approach can be used for movie post-production, gaming, helping artists reduce workload, generating synthetic data to develop machine learning techniques, etc. Note that our approach is not biased towards any specific gender, race, region, or social class. It works equally well irrespective of the difference in subjects.

However, the ability of generative models, including our approach, to synthesize images at a quality that some might find difficult to differentiate from source images raises essential concerns about different forms of disinformation, such as generating fake images or videos. Therefore, we believe the image synthesized using our approach must present itself as synthetic. We also believe it is essential to develop appropriate privacy-preserving techniques and large-scale authenticity assessment, such as fingerprinting, forensics, and other verification techniques to identify synthesized images. Such safeguarding measures would reduce the potential for misuse.

We also hope that the high-quality images produced by our approach could foster the development of the forgery mentioned above detection and verification systems. Finally, we believe that a robust public conversation is essential to creating a set of appropriate regulations and laws that would alleviate the risks of misusing these techniques while promoting their positive effects on technology development.

## REPRODUCIBILITY STATEMENT

We assure that all the results shown in the paper and supplemental materials can be reproduced. Furthermore, we will open-source our code together with pre-trained checkpoints. To reproduce our results, we provide the preprocessing procedures of each dataset and implementation details in the main paper and Appendix.

## ACKNOWLEDGEMENTS

Christian Theobalt was supported by ERC Consolidator Grant 4DReply (770784). Lingjie Liu was supported by Lise Meitner Postdoctoral Fellowship.

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

# A  ADDITIONAL MODEL DETAILS

## A.1  IMPLICIT FIELDS MODELING

**Modeling backgournd with NeRF++**  Most unstructured images used in this work have a complex and unbounded background, especially for the images taken from 360 degree directions (e.g., images in the CompCars dataset). Therefore, it is inefficient and difficult to model the entire embedded scene in the image within a fixed bounding box. Instead, following NeRF++ (Zhang et al., 2020), we partition the whole scene into foreground and background where the background is modeled with an additional network that takes as inputs the background latent codes and a 3D point $\boldsymbol{x} = (x, y, z)$ transformed using inverted sphere parameterization:

$$\boldsymbol{x}' = (x/r, y/r, z/r, 1/r), \ \text{ where } \ r = \sqrt{x^2 + y^2 + z^2} \tag{10}$$

Then, we uniformly sample $G$ background points in an inverse depth range: $[1/R, 0)$ where $R = 2.0$ in our experiments to represent where the background starts.

**Hierarchical volume sampling**  Similar to NeRF (Mildenhall et al., 2020), we adopt an efficient hierarchical sampling strategy for the foreground network learning. We first uniformly sample $N$ points between the near and far planes and then perform importance sampling of $M$ points based on the estimated density distribution in the coarse sampling stage. Different from NeRF (Mildenhall et al., 2020), we use a single network for coarse and fine sampling to predict the color and density.

## A.2  CAMERA

**Camera setup**  We assume that each image is rendered by a camera with a fixed intrinsic matrix in a pose sampled from a pre-defined camera pose distribution. The intrinsics are determined by field-of-view (FOV) and image resolution, normalized to $[-1, 1]$ regardless of the actual resolution. The camera is located on the unit sphere, pointing at the world origin where the target object is located. Then the camera pose $\boldsymbol{p}(\theta, \phi)$ is the function of pitch ($\theta$) and yaw ($\phi$). Typically, the cameras in real data do not follow the simple distributions that we assumed, which is the main problem for all 3D-aware GANs (including the proposed model) to learn the correct geometry. To tackle this issue, Niemeyer & Geiger (2021a) proposed to learn a neural network to generate camera simultaneously poses jointly with scene representations. However, our initial attempts of applying similar methods to *StyleNeRF* did not work, and the camera generator quickly diverged. Instead, we manually set the parameters $\phi$ and $\theta$ from either a Gaussian or uniform distribution depending on the datasets.

**Camera predictor**  Once *StyleNeRF* is trained, we can optionally train an additional camera predictor in a self-supervised manner. More specifically, we first randomly sample a camera pose and the latent codes to render the output image. Then, a CNN-based encoder (backbone initialized with a pretrained ResNet-18) reads the image and predicts the input camera parameters based on a smoothed L1 loss function. The camera predictor can be directly used on natural images, in particular for style inversion tasks (see Figure 7) as our early exploration shows that it is non-trivial to train the camera pose from random through latent optimization.

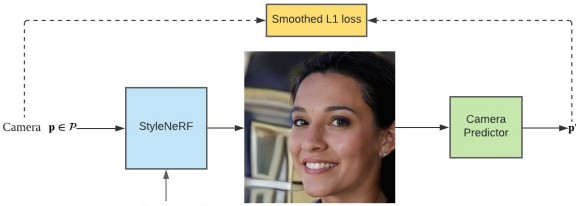

Figure 9: An illustration of self-supervised training the camera predictor.

## A.3  DERIVATION OF APPROXIMATION

The following shows the detailed derivation of Equation (6). For simplicity, we omit the condition on style vectors $\boldsymbol{\omega}$ in the equation.

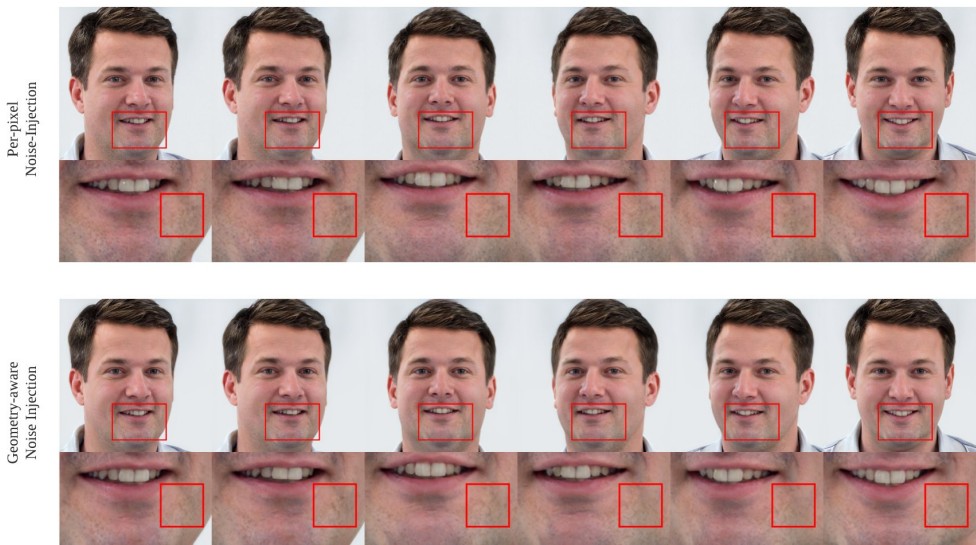

Figure 10: Example of sticking texture because of the injection of 2D noise at the test time. Note that the texture cropped inside the red box does not change across different views.

$$
\begin{aligned}
I^{\text{Approx}}(\boldsymbol{r}) &= \int_0^\infty p(t) \cdot h_c \circ [\phi^{n_c}(\boldsymbol{r}(t)), \zeta(\boldsymbol{d})] \, dt \\
&= \mathbb{E}_t \left( h_c \circ [\phi^{n_c}(\boldsymbol{r}(t)), \zeta(\boldsymbol{d})] \right) \\
&\approx h_c \circ \mathbb{E}_t \left( [\phi^{n_c}(\boldsymbol{r}(t)), \zeta(\boldsymbol{d})] \right) \\
&= h_c \circ [\mathbb{E}_t \left( \phi^{n_c}(\boldsymbol{r}(t)) \right), \zeta(\boldsymbol{d})] \\
&= h_c \circ [\mathbb{E}_t \left( \phi^{n_c,n_\sigma} \circ \phi^{n_\sigma}(\boldsymbol{r}(t)) \right), \zeta(\boldsymbol{d})] \\
&\approx h_c \circ [\phi^{n_c,n_\sigma} \circ \mathbb{E}_t \left( \phi^{n_\sigma}(\boldsymbol{r}(t)) \right), \zeta(\boldsymbol{d})] \\
&= h_c \circ \left[ \phi^{n_c,n_\sigma} \circ \int_0^\infty p(t) \cdot \phi^{n_\sigma}(\boldsymbol{r}(t)) dt, \zeta(\boldsymbol{d}) \right] \\
&= h_c \circ [\phi^{n_c,n_\sigma}(\mathcal{A}(\boldsymbol{r})), \zeta(\boldsymbol{d})]
\end{aligned}
$$

In the above derivation, we treat the volume rendering integral as an expectation. Therefore, if $p(t)$ is close to an impulse function which is near zero everywhere outside a small region near the object surface, locally $h_c$ and $\phi^{n_c,n_\sigma}$ can be seen as linear functions. Because the expectation of a linear function in an interval is equal to the value of the function evaluated at the middle point of the interval, the accuracy of the proposed approximation is well justified under this assumption.

### A.4 DETAILS ABOUT THE NOISE INJECTION

As mentioned in the paper, injecting per-pixel noise improves the generation quality of GANs. However, naively adding such noise as in Karras et al. (2019) is not feasible for our purpose because such noise exists in the pixel space, and it does not move when the viewpoint changes. On the other hand, the output gets noisy if we inject random noise for each frame separately. To avoid such inconsistency, the default setup in our experiments is to remove the noise injection as similarly done in (Karras et al., 2021) which works reasonably but has less ability to model local stochastic variations, resulting in lower visual quality (in terms of metrics like FID, KID, etc).

As an alternative, to improve synthesizing quality and keep multi-view consistency, we design a geometry-aware method for noise injection. Precisely, for each feature layer at the resolution of $N^2$, we extract the underlying geometry using marching-cube based on the predicted density. Next, we set the volume resolution $N^3$ accordingly. Then, we assign independent Gaussian noise at each vertex of the extracted mesh and render the noise map via rasterization from the same viewpoint. Finally, we inject the resulting noise to each layer before non-linear activation. Since this process

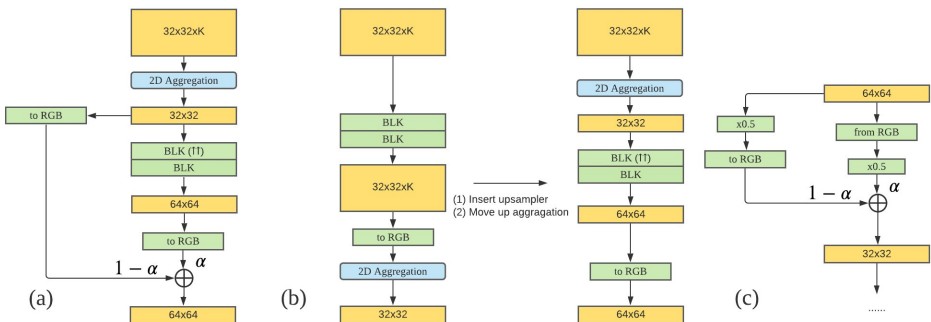

Figure 11: Illustrations of the proposed two variants of progressive training for *StyleNeRF*. (a) is for progressive growing of the generator; (b) is the same generator by progressively inserting upsamplers; (c) is the corresponding discriminator.

requires a proper density function to start with, we keep the standard per-pixel noise unchanged during training and only apply such geometry-aware noise at inference time. Compared to models without noise injection, although the proposed approach can capture more stochastic details, it is relatively slower (due to multi-scale marching-cube and rasterization), and the quality of the learned geometry constrains the final output.

### A.5 PROGRESSIVE TRAINING

As described in the main paper, it is essential to train *StyleNeRF* progressively from low to high resolutions to learn good geometry. We consider two types of progressive training:

**Progressive Growing** Following Karras et al. (2017), we start from a shallow network for low-resolution, and progressively fade in layers. Since the parameters of the newly added layers are random, it is essential to have a linear interpolation with low-resolution output to stabilize training.

**Progressive Up-sampling** Instead of growing new layers, another option is to train *StyleNeRF* by progressively inserting `upsample` operations during training as similarly done in Chan et al. (2021). In this case, we do not need to fade in layers linearly.

In both cases, the discriminator correspondingly grows from low to high resolutions progressively as described in Karras et al. (2017). Figure 11 illustrates progressive training for the generator and the discriminator. In practice, we found that both methods worked similarly in terms of visual quality, while progressive growing achieves a better speed advantage for low-resolution images. Therefore, our results are reported by training with progressive growing.

### A.6 HYPERPARAMETERS

We reuse the same architecture and parameters of StyleGAN2 for the mapping network (8 fully connected layers, $100\times$ lower learning rate) and discriminator. In addition, both the latent and style dimensions are set to $512$. For the foreground and background fields, before predicting the density, we use the style-based MLPs with $256$ and $128$ hidden units, respectively. The Fourier feature dimension is set to $L = 10$ (Equation (1)) for both fields. For layers after 2D aggregation, we follow the flexible layer configuration used by StyleGAN2, which decreases the feature sizes from $512$ (for $32^2$) until 32 (for $1024^2$). We keep most of the training configurations unchanged from StyleGAN2.

### A.7 IMPLEMENTATION

We implement our model based on the official Pytorch implementation of StyleGAN2-ADA [1]. By default, we train the *StyleNeRF* model by going through $25000k$ images with a minibatch $= 64$. All models are trained on 8 Tesla V100 GPUs for about three days.

---

[1]https://github.com/NVlabs/stylegan2-ada-pytorch

Table 2: (*Left*) FID, KID $\times 10^3$ on images with $512^2$ and $1024^2$ pixels. 2D GAN represents the StyleGAN2 (Karras et al., 2020b). (*Right*) We also compare the results of replacing $3 \times 3$ `Conv` with $1 \times 1$ and/or removing per-pixel noise injection for 2D models. We use **bold** font to represent the default setting for both our model.

| Models | FFHQ $512^2$ | | AFHQ $512^2$ | | MetFace $512^2$ | | FFHQ $1024^2$ | |
|---|---|---|---|---|---|---|---|---|
| | FID | KID | FID | KID | FID | KID | FID | KID |
| 2D GAN | 3.1 | 0.7 | 8.6 | 1.7 | 18.9 | 2.7 | 2.7 | 0.5 |
| Ours | 7.8 | 2.2 | 13.2 | 3.6 | 20.4 | 3.3 | 8.1 | 2.4 |

| $1 \times 1$ | Noise | 2D GAN | Ours |
|---|---|---|---|
| Yes | No | 11.4 | **7.8** |
| Yes | Yes | 5.9 | 6.0 |
| No | No | 4.6 | - |

## B  DATASET DETAILS

**FFHQ** (`https://github.com/NVlabs/ffhq-dataset`) contains 70k images of real human faces in resolution of $1024^2$. We assume the human face to be captured at the origin. In the training stage, we sample the pitch and yaw of the camera from the Gaussian distribution.

**AFHQ** (`https://github.com/clovaai/stargan-v2#animal-faces-hq-dataset-afhq`) contains in total 15k images of animal faces including cat, dog and wild three categories in resolution of $512^2$. We directly merge all training images without using the label information. Similar to the FFHQ setting, we sample the pitch and yaw from the Gaussian distribution.

**MetFaces** (`https://github.com/NVlabs/metfaces-dataset`) contains 1336 images of faces extracted from artworks. The original resolution is $1024^2$, and we resize it to $512^2$ for our main experiments. As mentioned in the paper, we finetune *StyleNeRF* from the FFHQ checkpoint due to the small size of MetFaces.

**CompCars** (`http://mmlab.ie.cuhk.edu.hk/datasets/comp_cars/`) contains 136726 images capturing the entire cars with different styles. The original dataset contains images with different aspect ratios. We preprocess the dataset by center cropping and resizing them into $256^2$. In the training stage, we sample the camera 360 degree uniformly.

## C  BASELINE DETAILS

**HoloGAN (Nguyen-Phuoc et al., 2019)**  We train HoloGAN on top of the official implementation [2]. Specifically, HoloGAN is trained in resolution $256^2$ with Adam optimizer with an initial learning rate of 5e-5. We train each HoloGAN model for 50 epochs. The learning rate linearly decays between epoch 25 and 50. As original HoloGAN only supports training on images of resolution $64^2$ and $128^2$, to synthesize images of resolution $256^2$ we follow the same adaptation scheme as used in GIRAFFE (Niemeyer & Geiger, 2021b) that add one convolution layer with AdaIn (Huang & Belongie, 2017) and leaky ReLU activation on top of the official synthesis network.

**GRAF (Schwarz et al., 2020)**  We adopt the official implementation [3] and retrain GRAF models on FFHQ, AFHQ and preprocessed CompCars dataset in resolution $256^2$. Following the default setting, all models are trained at the target resolution using the patch-based discriminator.

**$\pi$-GAN (Chan et al., 2021)**  Same as GRAF, we use the official implementation [4] and retrain $\pi$-GAN models on these three datasets: FFHQ, AFHQ and CompCars. Due to the high cost of running $\pi$-GAN in high-resolution, we follow the same pipeline, which progressively increases the resolution from $32^2$ until $128^2$, and we render the final outputs in $256^2$ by sampling more pixels.

**GIRAFFE (Niemeyer & Geiger, 2021b)**  For FFHQ dataset, we directly use their pretrained models of resolution $256^2$. For AFHQ and CompCars dataset, we retrain GIRAFFE with the official Giraffe implementation [5]. We change the default random crop preprocessing to center crop to compare with other approaches.

---

[2]https://github.com/thunguyenphuoc/HoloGAN
[3]https://github.com/autonomousvision/graf
[4]https://github.com/marcoamonteiro/pi-GAN
[5]https://github.com/autonomousvision/giraffe

Table 3: PSNR, SSIM and LPIPS scores between the images synthesized by 3D-aware generative models, and the corresponding pseudo targets generated by IBRNet (Wang et al., 2021).

| Models | FFHQ $256^2$ | | | AFHQ $256^2$ | | |
|---|---|---|---|---|---|---|
| | PSNR ↑ | SSIM ↑ | LPIPS ↓ | PSNR ↑ | SSIM ↑ | LPIPS ↓ |
| $\pi$-GAN | 29.5 | 0.92 | 0.04 | 28.5 | 0.86 | 0.12 |
| GIRAFFE | 25.8 | 0.81 | 0.13 | 26.2 | 0.75 | 0.25 |
| Ours | 29.0 | 0.89 | 0.08 | 26.8 | 0.80 | 0.13 |

Table 4: FID on the outputs from *StyleNeRF* trained with different upsampling operators.

| Metric | Bilinear | LIIF | PixelShuffle w/o blur | Proposed |
|---|---|---|---|---|
| FID | 12.4 | 10.6 | 32.4 | **7.8** |

## D   ADDITIONAL RESULTS

**Comparison to StyleGAN2**   We also provide quantitative comparisons on high-resolution image synthesis against StyleGAN2 and a variant that adopts comparable settings with our default model. As shown in Table 2, there is still a gap to the best 2D-GAN models. Nevertheless, the proposed *StyleNeRF* reaches similar or even better results as 2D models in similar architectures. This implies the quality drops are mainly from less powerful architecture (e.g., no noise injection, no $3 \times 3$ convolutions). As future work, more exploration can be done to close this gap.

**Quantitative comparison on 3D Consistency**   We conduct quantitative comparisons to measure the multi-view consistency of the synthesized images from *StyleNeRF* and two baselines ($\pi$-GAN, GIRAFFE). We do not employ the metric used in GRAF (Schwarz et al., 2020) because the calculation of that metric requires 3D ground-truth geometry, and thus it can only be applied to synthetic data, not real data where the ground-truth geometry is not available.

Instead, we propose a new method to measure multi-view consistency without the requirement of 3D shapes. First for each model, we randomly sample 1000 seeds, and for each seed, we generate a sequence of 9 images following the same trajectory. Next, we take four images as conditioning, and predict pseudo ground truth images given the remaining five camera poses using a pre-trained IBRNet (Wang et al., 2021), a recently proposed image-based rendering method for generic view interpolation and can generalize to novel scenes. Then we measure the typical image reconstruction scores (PSNR, SSIM, and LPIPS (Zhang et al., 2018)) between the pseudo targets and the remaining images. Since the IBRNet model is fixed for all the methods, the reconstruction scores can be used as a metric to measure the multi-view consistency of different methods (the better the reconstruction scores are, the higher consistency the synthesized images are).

We tested on the FFHQ $256^2$ and AFHQ $256^2$ and reported the results in Table 3. *StyleNeRF* achieves much better consistency than GIRAFFE, while it is comparable to $\pi$-GAN, the SoTA pure NeRF-based model, in terms of multi-view consistency. Meanwhile, *StyleNeRF* can generate high-resolution images efficiently, which cannot be achieved by existing pure NeRF-based models.

**3D Reconstruction**   We also include the COLMAP (Schönberger & Frahm, 2016) reconstruction example of the proposed *StyleNeRF* on FFHQ to validate the 3D consistency of the model's output. As shown in Figure 12, given a style, we sample 36 camera poses to generate images and obtain the reconstructed point clouds from COLMAP with default parameters and no known camera poses.

**Ablation on upsamplers**   We test different upsampling operators quantitatively on FFHQ $512^2$ and report the FID scores in Table 4. We also include the additional result of using pixelshuffle without applying the blur kernel as described in Equation (6). The FID results show that the proposed upsampler achieves the best quality compared to other operators.

**Additional Visual Results**   We show additional results of generating images with different camera views as follows. Please also refer to the supplemental video, which shows more results.

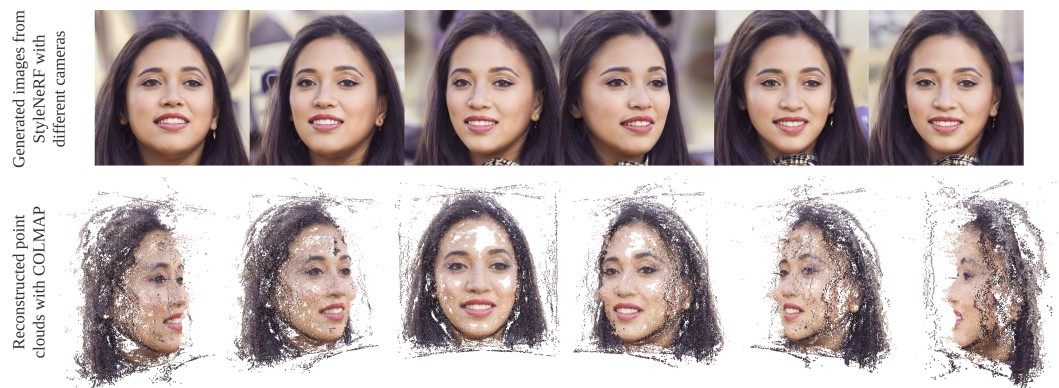

Figure 12: COLMAP reconstructions for models trained on FFHQ $512^2$

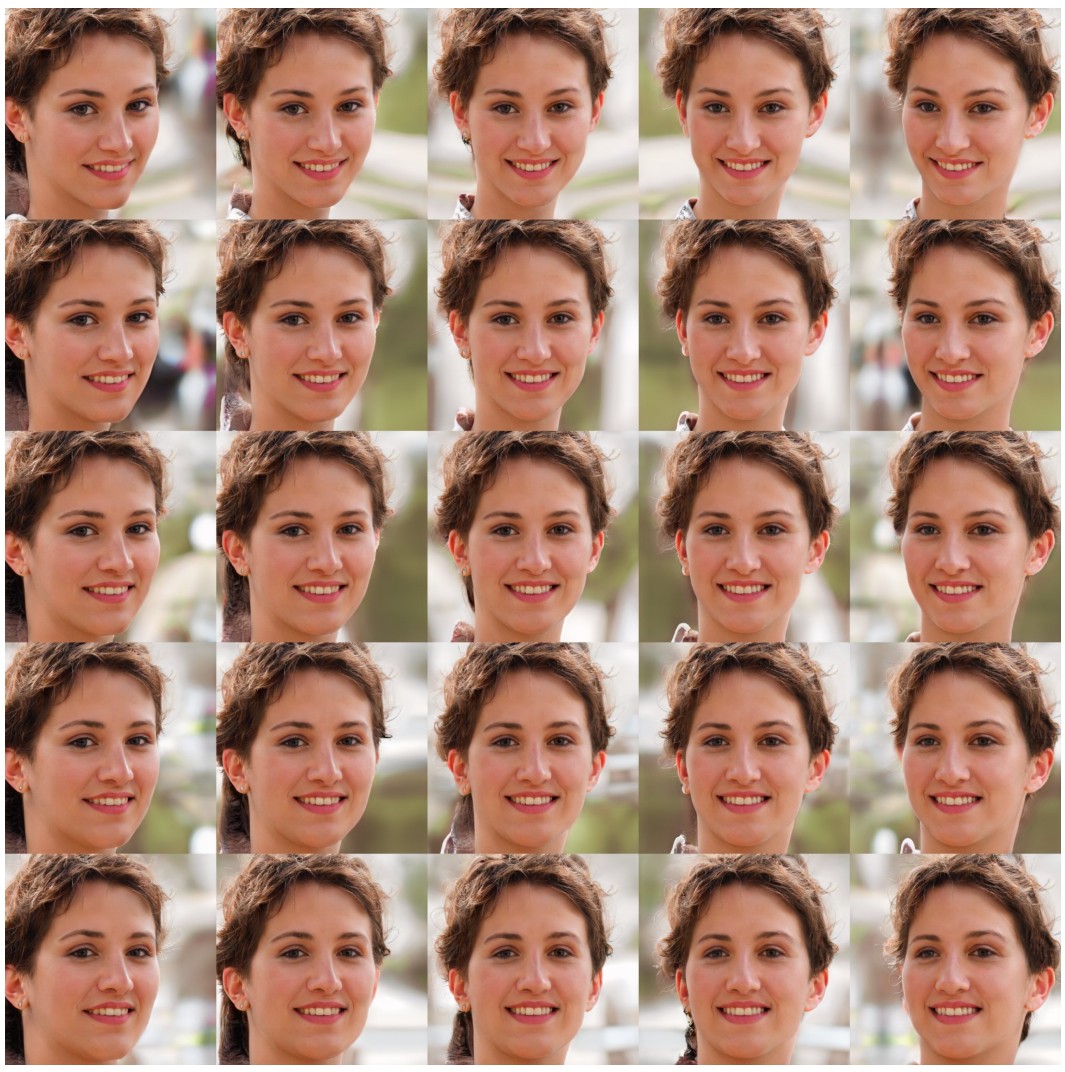

Figure 13: Example on FFHQ $1024^2$ displayed from different viewing angles.

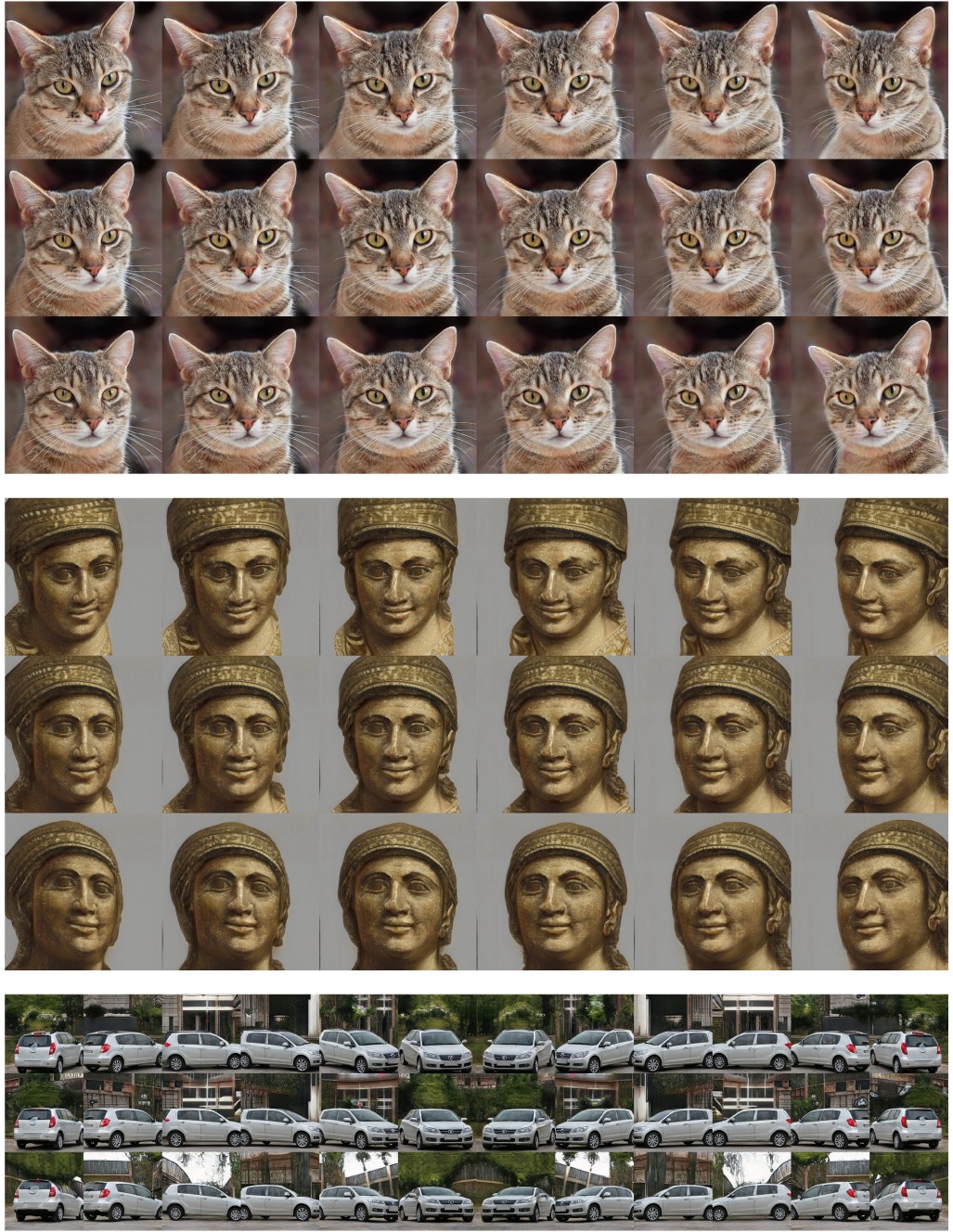

Figure 14: Example on AFHQ ($512^2$), MetFaces ($512^2$) and CompCars ($256^2$) displayed from different viewing angles.

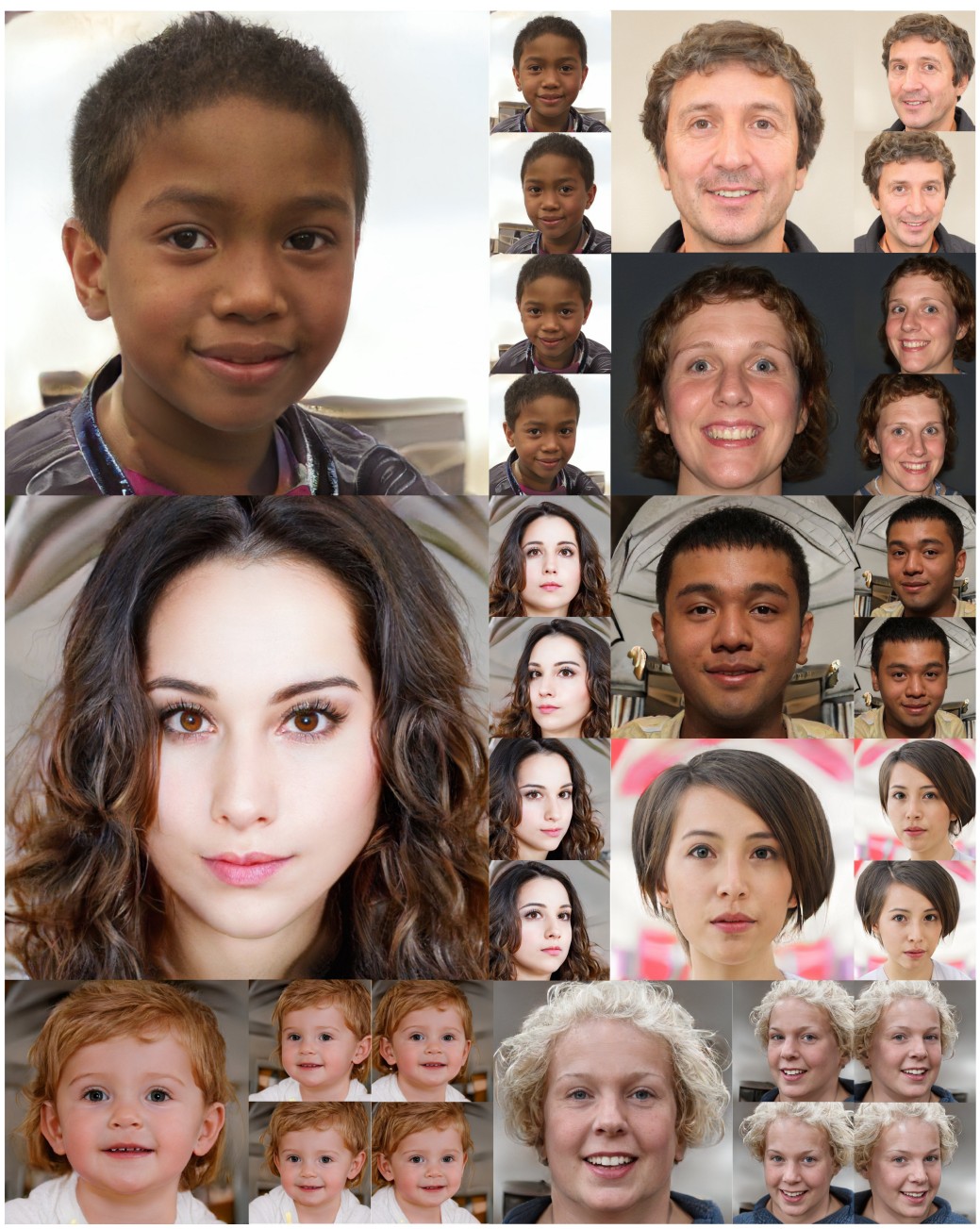

Figure 15: High resolution samples with explicit camera control on FFHQ $1024^2$.

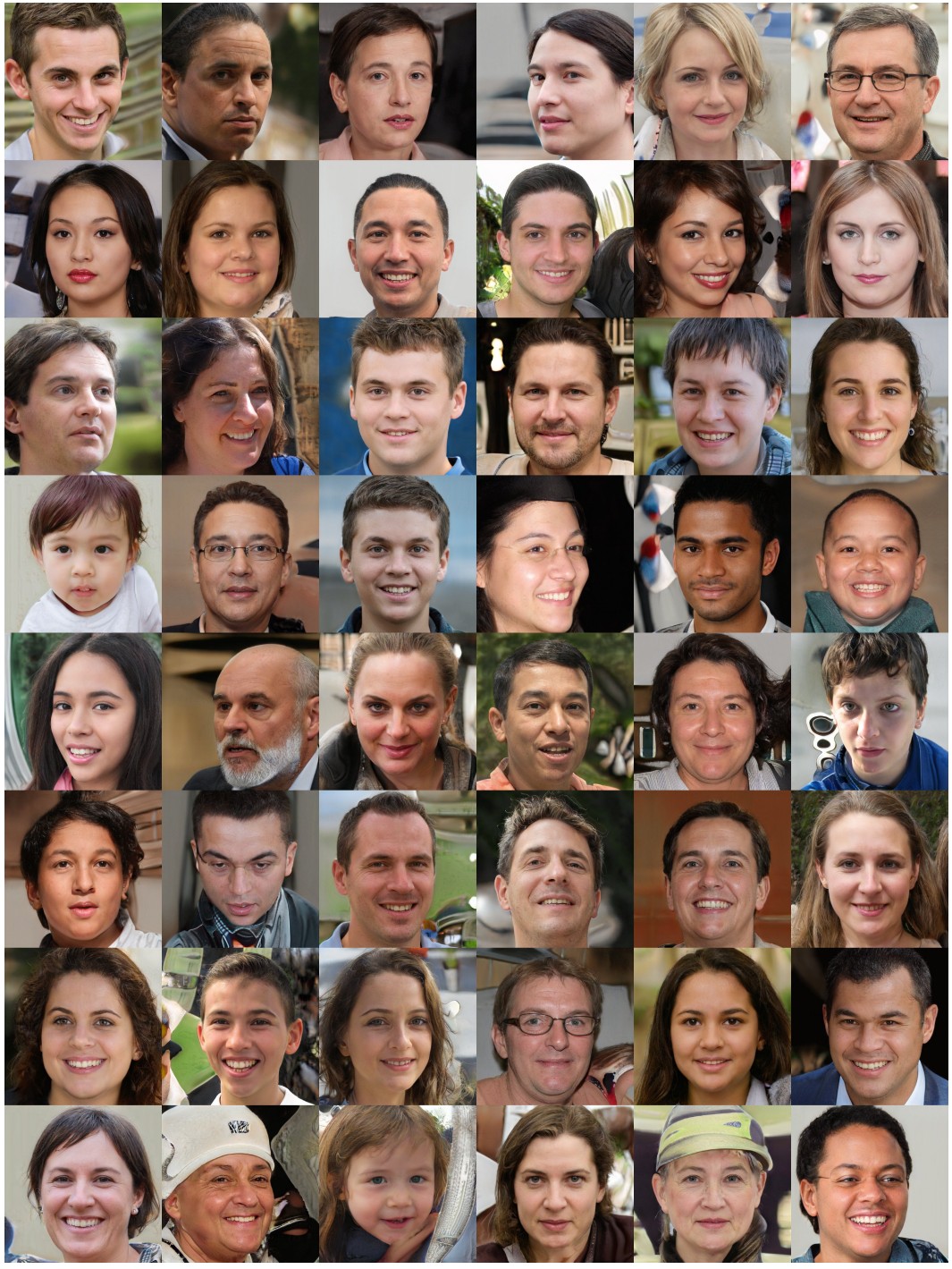

Figure 16: Random samples with random camera poses on FFHQ $512^2$.

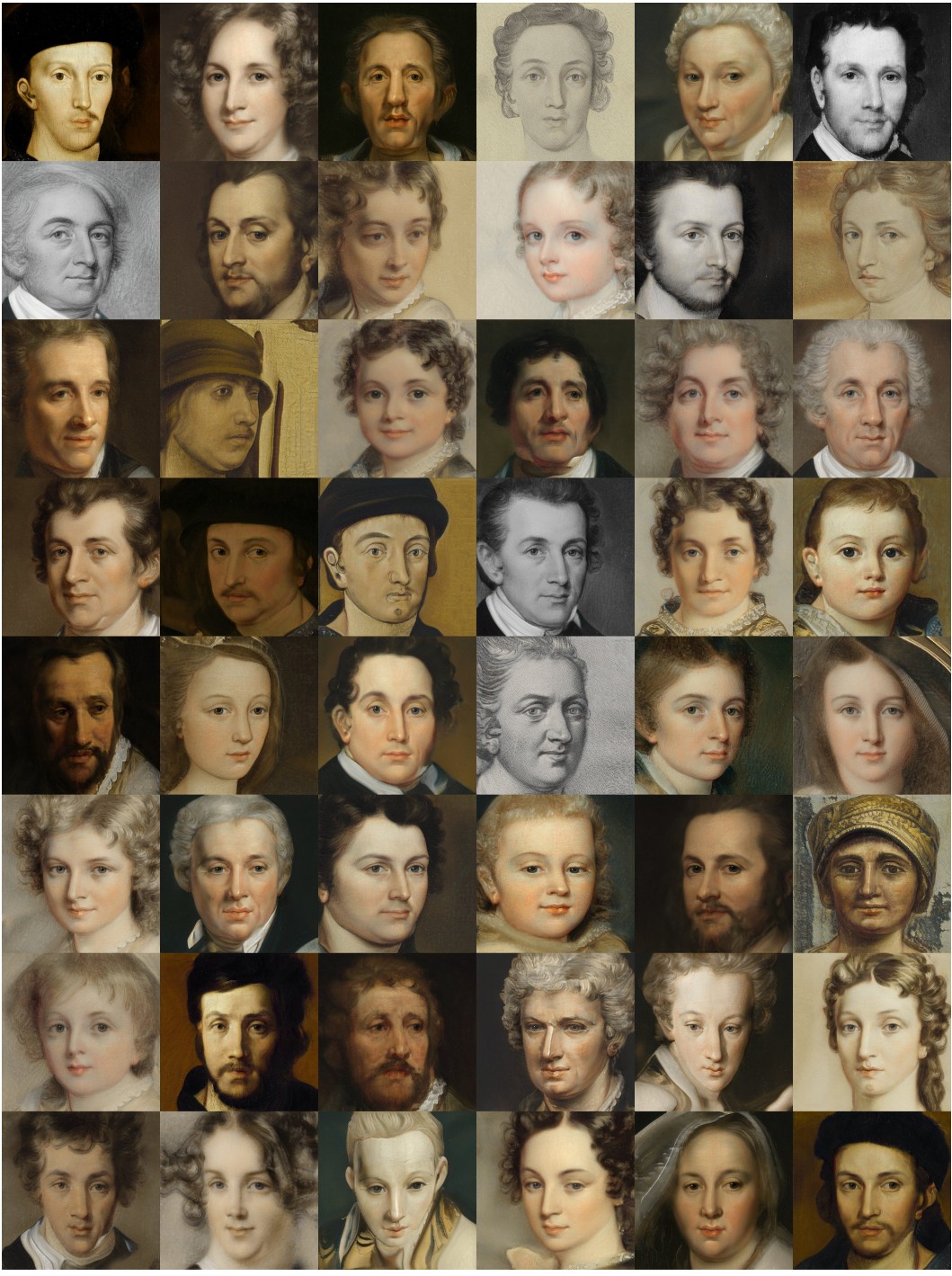

Figure 17: Random samples with random camera poses on MetFaces $512^2$.

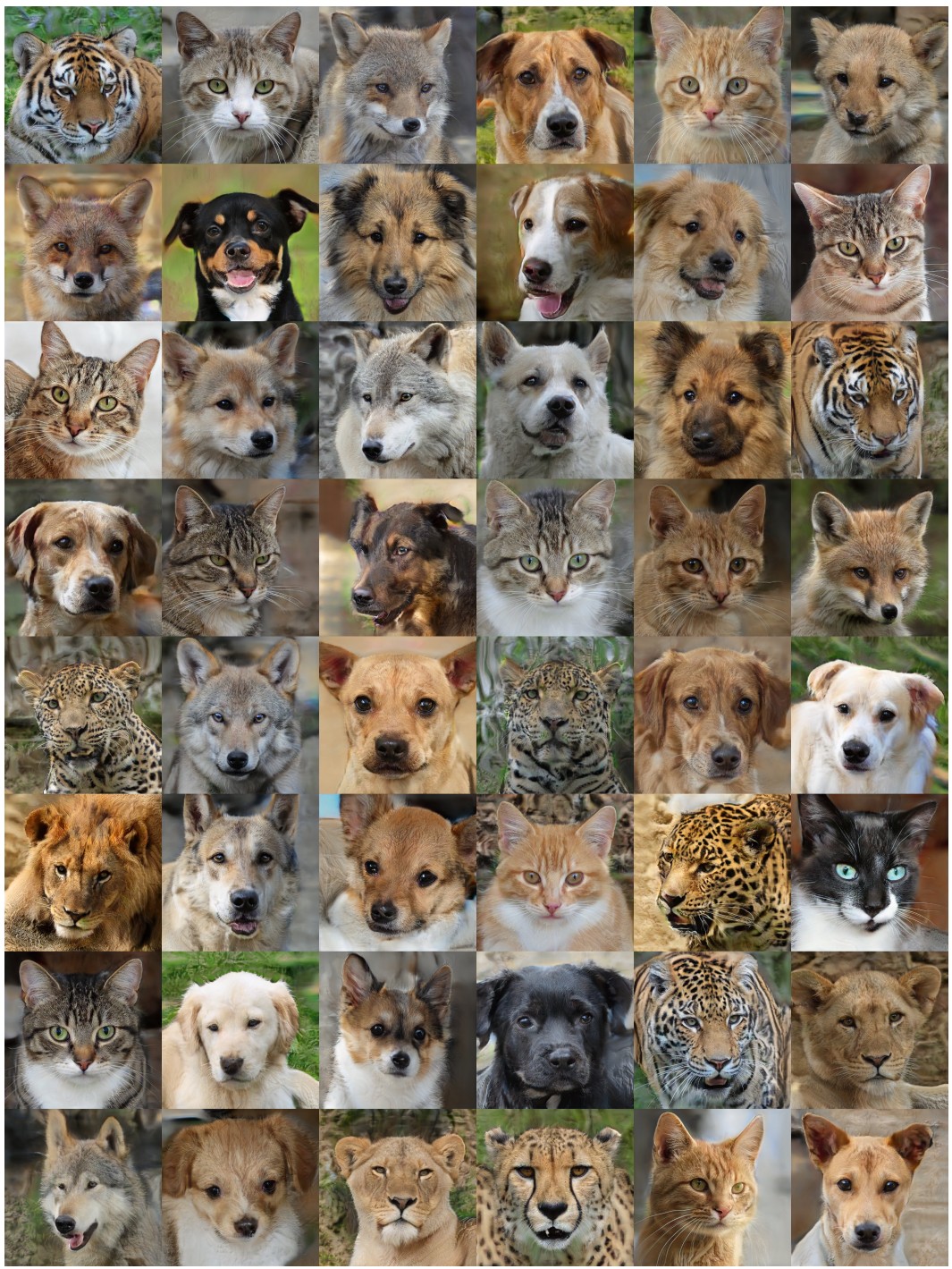

Figure 18: Random samples with random camera poses on AFHQ $512^2$.

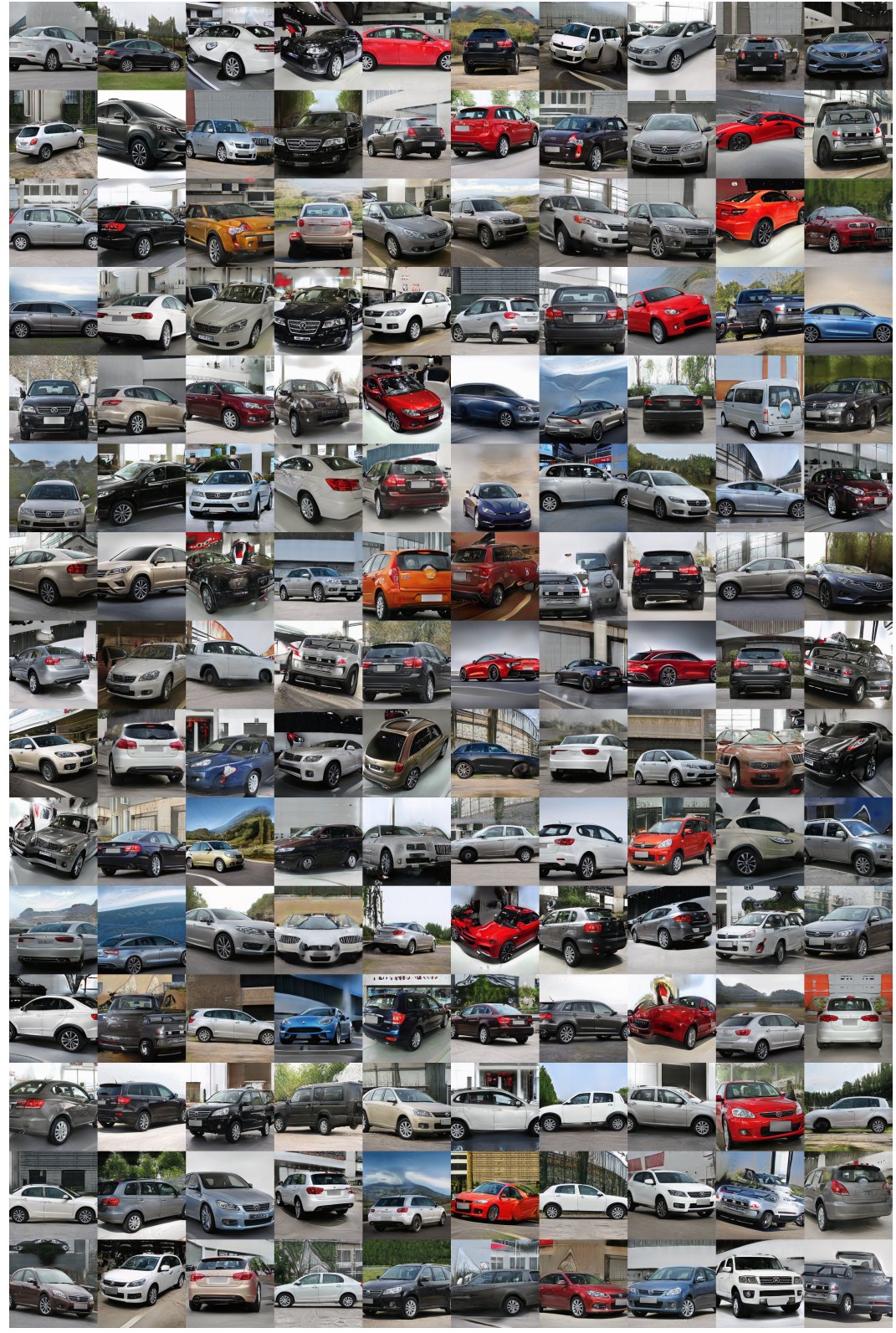

Figure 19: Random samples with random camera poses on CompCars $256^2$.

