# OpenReview forum: "StyleNeRF: A Style-based 3D Aware Generator for High-resolution Image Synthesis"
_ICLR.cc/2022/Conference — ICLR 2022 Poster_

### Official Review · Reviewer_MWb6 · 2021-10-29

**Correctness:** 4
**Technical Novelty And Significance:** 3
**Empirical Novelty And Significance:** 3
**Recommendation:** 8
**Confidence:** 5

**Main Review:**

Strengths:
(1) StyleNeRF gives better qualitative and quantitative results than the baseline in high resolution situations, which is particularly evident in the fewer artifacts. Also, this method adopts rendering at interactive rates. (2) The paper solves the problem of high-resolution synthesis that cannot be solved by traditional NeRF-based methods. (3) The proposed pioneering approach of fusing 2D GAN tricks and NeRF-based tricks is a meaningful and groundbreaking work.

Weaknesses:
(1) There are some typos. In Figure 3. The 'r' in 'right' should be 'R'. The 'an mapping network' should be 'a mapping network' in the third row of Section 3.4. The pi-GAN and π-GAN(in Baseline details of Appendix C) should be written in the same way. (2) Figure 3 is quite confusing, the blocks in figure should be mentioned in paper, like FG and BLK. (3) It is suggested to conduct more quantitative experiments about different upsampling operators to show the strengths of the proposed method. (4) There is an interesting phenomenon where the pupil position in the human eyes of the synthesized images (Figure 6 and 15)  are different so that the human eyes seem to toward the direction of the camera observation, while the pupil position in the human eyes of the synthesized images (Figure 12 and 13)  are nearly same. Could this be explained?


**Summary Of The Paper:**

This paper proposes StyleNeRF which combines NeRF and a style-based generator to improve rendering efficiency and 3D consistency in high-resolution image synthesis. In this paper, (1) The NeRF is used to produce a low-resolution feature map and upsample it progressively to high resolution. (2) Several designs are proposed to improve 3D consistency, including a desirable upsampler, a novel regularization term. (3) A progressive training strategy is adopted to significantly improves the stability of learning the real geometry.

**Summary Of The Review:**

 I think the contributions of this paper far outweigh its shortcomings and this paper can be accepted.

---

> ### Author Response · Authors · 2021-11-21
> **Response to Reviewer MWb6**
>
> 1. and 2. Thanks for pointing out the typos and unclear notations. Please see general response 2.3
>
> 3. We also tested different upsampling operators quantitatively on FFHQ 512x512, and reported the FID scores in the following table.
> From FID, we can see that the proposed upsampler achieves the best quality compared to other operators. We have added this experiment to the revision Appendix D.
>
> | Operator | FID |
> |:----------|:--------------:|
> | Bilinear|   12.4 |
> | LIIF | 10.6 |
> | Pixelshuffle w/o blur | 32.4 |
> | Our proposed | 7.8 |
>
>
>
> 4. Please check.general response 2.2

---

> > ### Comment · Reviewer_MWb6 · 2021-11-29
> > **Response Checked**
> >
> > Thanks very much! After checking the response and the comments from other reviewers, I decide to keep the original score.

---

### Official Review · Reviewer_gKfo · 2021-11-02

**Correctness:** 4
**Technical Novelty And Significance:** 2
**Empirical Novelty And Significance:** 3
**Recommendation:** 6
**Confidence:** 5

**Main Review:**

With the great success that has been achieved by StyleGAN in 2D, it is natural to extend the key idea behind StyleGAN to 3D for a 3D-aware generative model. It is clearly shown in the paper that this work has strengths including:
1. a very good empirical effort to extend StyleGAN to 3D.
2. the proposed method produces SOTA results, the generated images are impressive.
3. the proposed method is fast and supports high-resolution renderings.

Meanwhile, there are also several weaknesses:
1. lack of technical novelty, the proposed combines several existing techniques that are well explored in the GAN field.
2. the key contribution that makes this work conceptually different may be unclear. While it is true that all the tricks as a whole deliver impressive results, it is not easy to identify the unique contribution in this work.
3. 3D-aware generative models have a focus on multi-view consistency, in contrast to 2D GANs. However, the multi-view consistency (particularly at fine levels) is compromised when resorting to the CNN-based renderer to produce the final image. The proposed pipeline is a good workaround but this compromise is absolutely nonnegligible, it does not fundamentally solve the key issue existing in this 3D-aware generative method. This deviates from the original course of the 3D-aware generative model.

More that need clarification:
1. what is the geometric meaning of the equation (5)? Is it that only the feature at the distance of the depth value is hit for the subsequent rendering? If this is the case, the rendering process still has to evaluate the density of all the samples along the ray for obtaining the depth value. I am not sure about the gain of this approximation in terms of improving computation efficiency. Would appreciate it if this can be explained.
2. how general is the upsampler design in this work? It seems to work very well in the paper. Would be exciting to see if this design choice can also be used in other GANs to improve the synthesis and thus show the generalizability of this design choice.
3. this work uses separate networks for the background and the foreground (based on NeRF++), I am wondering if this is critical, as this design is not employed in other baseline methods, in which the generated images have their bg and fg twisted together.
4. it is a bit confusing that GRAF and pi-GAN perform so much worse than in their original paper. The datasets used in this paper are different from theirs, comparisons on their datasets may be a better option to rule out those irrelevant factors (e.g., hyper-parameters) and thus offer a fairly understandable comparison.
5. in Table 1, I understand that FID and KID do not penalize the 2D GAN for having no multi-view consistency, but still, there should be proper highlights in this table... We can have a metric for quantitatively measuring the multi-view consistency (e.g., like in GRAF, measuring the multi-view reconstruction quality of generated multi-view images of the same 3D), on which the proposed method would dominate.

Others:
1. section 3.2, around equation (5), some of the Phi may have incorrect superscripts, causing some troubles in following this part.
2. section 3.4: "an mapping network" -> "a mapping network".
3. it may be better to not expand the equation (2) and not expose many g, the g functions are not used elsewhere, so it may only make it lengthy and a bit more unreadable.
4. section 4.2: "GARF" -> "GRAF".

**Summary Of The Paper:**

This paper proposes a 3D-aware generative model for photo-realistic high-resolution image synthesis. The core of this work is a set of tricks that are employed to speed up the generation and enforce the multi-view consistency while trying to synthesize higher resolution images in 2D with a CNN-based renderer.

**Summary Of The Review:**

Overall, I appreciate very much the engineering effort behind this work. Although the lack of significant conceptual novelty, it has truly brought up the 3D-aware generative model onto a new level with great quality. Please check the concerns expressed above for not giving a higher rating.

---

> ### Author Response · Authors · 2021-11-21
> **Response to Reviewer gKfo**
>
> ### Responses to weaknesses
> 1. Please see general response 1.1
>
> 2. While there is no single technique that itself can be summarized to solve the entire problem, we believe the goal to achieve is more important than a single trick that works magically. The significant contribution of this work is that, for the first time, a 3D-aware  generative model can efficiently synthesize high-resolution images with high quality and multi-view consistency.All the proposed techniques contribute to this goal.
>
> 3. We agree that our work does not entirely solve the perfect multi-view consistency issue in 3D-aware generative models for high-resolution image synthesis. However, we believe that our work is a significant step in the right direction (also acknowledged in the public review from Miguel A Bautista). To approach this challenging problem, we proposed several novel strategies (e.g., upsampler design, a new NeRF path regularizer, removing view direction and progressive training) that enable high-resolution image synthesis at interactive rates while preserving high multi-view consistency. Our method also outperforms existing 3D-aware models in synthesis quality with a significant margin. We believe this work would inspire more theoretical studies in preserving perfect multi-view consistency in high-resolution image synthesis.
>
> ### Responses to clarification
> 1. Yes, we still need to evaluate the densities for each sampled point, but we are able to evaluate the color for each ray rather than for each sampled point. Furthermore, it enables the use of upsampling for further approximation to save computation time.
>
> 2. Yes, we tested different upsamplers in 2D baselines and found that the designed upsampler brings significant improvements only when the convolution kernel size is 1x1. For standard 2D models based on convolution with kernel size > 1, the proposed upsampler does not make a big difference.
>
> 3. GIRAFFE also has separate networks for foreground and background; other baselines do not. We also tried our model without using two separate networks, and the final results are similar. In our current model, we chose the two separate networks design because it makes more sense to represent 3D scenes with a separate background .
>
> 4. Please see general response 1.2
>
> 5. Please see general response 2.1

---

### Official Review · Reviewer_TLAR · 2021-11-02

**Correctness:** 2
**Technical Novelty And Significance:** 2
**Empirical Novelty And Significance:** 2
**Recommendation:** 6
**Confidence:** 4

**Main Review:**

Strengths:
+ The proposed method is able to synthesize high-resolution images at interactive rates while preserving 3D consistency at high quality.
+ It enables control of camera poses and different levels of styles.
+ It also supports new tasks such as style mixing, inversion, and simple semantic edits.

Weaknesses:
- The paper lacks novelty. The method is mostly a simple combination of Pi-GAN which uses a StyleGAN-alike MLP for NeRF, and GIRAFFE which uses volume rendering to generate low-resolution features that are sequentially upsampled by CNN. Thus, the contribution of this work seems not particularly strong.
- The approximation relationship in Eq. 5 is not quite obvious. Please provide more detailed proof and explanations for it.
- In Figure 2, it is not clear how the internal representations are visualized? What do different colors mean? In addition, the mentioned artifacts can hardly be noticed in the final image output.
- The upsampler design is not quite clear. It is hard to understand how such a design solves the “chessboard”, “texture sticking” and "bubble" artifacts.
- What is $\varphi_\theta$ in Eq. 7?
- For evaluation, in Figure 4, the results of the baselines, e.g pi-GAN, are surprisingly bad. This is not consistent with the results in the original paper. This requires some explanations.
- In Figure 3, what is $K_{fg}$ and $K_{bg}$, and how to obtain them?

**Summary Of The Paper:**

The paper presented StyleNeRF, a 3D-aware generative model for high-resolution image synthesis with high multi-view consistency.
StyleNeRF integrates the neural radiance field (NeRF) into a style-based generator to improve rendering efficiency and 3D consistency.
It performs volume rendering only to produce a low-resolution feature map, and progressively applies upsampling in 2D.
It also presents a new upsampling module and a new regularization loss to enforce 3D consistency.


**Summary Of The Review:**

This is a borderline paper.  It achieves very interesting results. However, it has quite a few issues in terms of novelty and evaluation, and many algorithm designs and technical details are not well explained.

---

> ### Author Response · Authors · 2021-11-21
> **Response to Reviewer TLAR**
>
> 1. Please see general response 1.1
> 2. The following shows the detailed derivation of Eq. 5. In the derivation, we assume that $p(t)$ is close to an impulse function which is near zero everywhere outside a small region near the object surface. Hence, locally the functions $h_c$ and $\phi^{n_c,n_\sigma}$ can be seen as linear functions. Because the expectation (mean) of a linear function in an interval is equal to the value of the function evaluated at the middle point of the interval, the accuracy of the proposed approximation is well justified under this assumption. We also added this derivation in the revision Appendix A.3
>
> $
> I(r)=\int_0^\infty p(t)\cdot h_c \circ \left[\phi^{n_c}(r(t)),\zeta(d)\right]dt
> $
>
> $
> \ \ \ \ \ \ \ = \mathbb{E}_t \left(h_c \circ \left[\phi^{n_c}(r(t)),\zeta(d)\right]\right)
> $
>
> $
> \ \ \ \ \ \ \ \approx  h_c \circ \mathbb{E}_t \left(\left[\phi^{n_c}(r(t)),\zeta(d)\right]\right)
> $
>
> $
> \ \ \ \ \ \ \ =  h_c \circ \left[ \mathbb{E}_t\left(\phi^{n_c}(r(t))\right),\zeta(d)\right]
> $
>
> $
> \ \ \ \ \ \ \ =  h_c \circ \left[ \mathbb{E}_t\left(\phi^{n_c,n_\sigma}\circ\phi^{n_\sigma}(r(t))\right),\zeta(d)\right]
> $
>
> $
> \ \ \ \ \ \ \ \approx  h_c \circ \left[\phi^{n_c,n_\sigma}\circ\mathbb{E}_t\left( \phi^{n_\sigma}(r(t))\right),\zeta(d)\right]
> $
>
> $
> \ \ \ \ \ \ \ =h_c \circ \left[\phi^{n_c,n_\sigma}\circ\int_0^\infty p(t)\cdot \phi^{n_\sigma}(r(t))dt,\zeta(d)\right]
> $
>
> $
> \ \ \ \ \ \ \ = h_c \circ \left[ \phi^{n_c,n_\sigma}(\mathcal{A}(r)),\zeta(d)\right]
> $
>
>
> 3. We followed the visualization of feature images used in StyleGAN3 (Karras et al., 2021). The internal representations are three selected channels of the feature maps at different levels. Actually, the artifacts in the final images can be easily observed in a lot of other samples (The sample in Fig 2 seems not that obvious due to the small color contrast on skins). We have replaced an example for better visualizing artifacts in the revised paper and Supplementary video.
>
> 4. First of all, the choice of upsampler was empirical, and we gave our insight on why the selected upsampler migrates the mentioned artifacts in Sec. 3.2. We first tested typical upsampling functions used in GAN literature, which we summarized into two categories: “fixed filter” (e.g., bilinear, bicubic, or FIR filters) and “pixel-wise learnable transformation” (e.g., pixelshuffle, LIIF, ConvTranspose). Although both cases work similarly in 2D GANs with standard 3x3 Conv, they fall short in our case when only MLPs (1x1 Conv) are used. In our view, the former enables smooth and symmetric interpolation weights for each location, which tends to produce “bubble shape” output; the latter avoids the above issue by learning to upsample differently at each direction. However, it is not smooth as it lacks information about surrounding pixels. Therefore, we propose to combine these two categories for the best performance. More theoretical analysis can be made as to the future work, as discussed in Sec. 4.5.
>
> 5. As written on page 5 of the paper, $\psi_\theta$ is a learnable 2-layer MLP used before pixelshuffle.
>
> 6. Please see general response 1.2
>
> 7. Please see general response 2.3

---

### Official Review · Reviewer_Qm5b · 2021-11-03

**Correctness:** 4
**Technical Novelty And Significance:** 3
**Empirical Novelty And Significance:** 3
**Recommendation:** 10
**Confidence:** 3

**Main Review:**

Strength:
(1) The proposed StyleNerF Model can synthesize high resolution, photo-realistic images with strong multi-view consistency, while achieving interactive rendering rate. Furthermore, it allows controls of camera poses and style attributes.
(2) To tackle the problem of efficiency, authors first use NeRF to volume render low resolution feature maps, and then feed into stylegan 2D synthesis blocks to synthesize high resolution images. To avoid multi-view inconsistencies, authors proposed to use NeRF path regularization along with other design choices.
(3) Extensive experiments on several benchmarks as well as ablation study validate the effectiveness of proposed component and demonstrate state of the art results on 3D generative view synthesis.

Weakness:
(1) The proposed method remove viewing direction from the NeRF to avoid ambiguity learned from single images, which is fine, but strictly speaking it’s no longer a “radiance field”, but instead a “irradiance fields”.
(2) In Figure 3, authors need to add clarification for each notation (such as FG,BG, BLK) to make reader easier for readers to understand.
(3) From the demo video, I saw a “gaze following” phenomena, where the eye looking direction always follows the camera path, is there any explanation and way to fix this problem?
(4) I am curious if the proposed method can also synthesize 3D scenes from unstructured photos. For example, StyleGAN is able to train on photos of nature landscapes to generate beautiful 2D images. I am wondering what would happen if the proposed model was train on such data.


**Summary Of The Paper:**

In this manuscript, authors proposed a novel 3D-aware generative model for photo-realistic high resolution image synthesis. The proposed method combines Neural Radiance Fields (NerF) and StyleGAN, and tackles the challenges of efficiency, multi-view consistency and rendering quality. Specifically, authors propose to use NeRF to render low resolution images before feeding them into 2D StyleGAN network to upsample the feature maps in order to obtain images with high resolution and therefore the proposed model can render images at interactive rates. Extensive experiments on numerous datasets demonstrate superior performance than prior state of art 3D generative view synthesis approaches and largely close the gap between 2D and 3D view synthesis methods.

**Summary Of The Review:**

Based on the strength and weakness I mentioned above, I strongly recommend this paper to be accepted because its novelty on closing the gap between 3D GAN and 2D GAN problem and this is the first 3D generative view synthesis work that can render high resolution images with interactive speed while preserving very high rendering quality. Its proposed component can be very useful to push the boundaries of this area in the future.

---

> ### Author Response · Authors · 2021-11-21
> **Response to Reviewer Qm5b**
>
> 1. Thanks for your comments. In radiometry, “irradiance” is the radiant flux (power) received by a surface per unit area from all possible incoming directions, while “radiance” is the power received or emitted by a unit area from a direction. Our method models the emitted power from a direction at any 3D point. Therefore we use the term “radiance” rather than received power, i.e., “irradiance.” Note that omitting the view direction input makes the emitted radiance invariant across different view directions, which amounts to assuming that the object surface is perfectly diffuse.
>
> 2. Please see general response 2.3
>
> 3. Please see general response 2.2
>
> 4. Good point. One limitation of the current StyleNeRF is that it only works well on the datasets where objects are roughly aligned at the coordinate center with camera viewpoints approximately in distribution we assume (e.g., Gaussian or uniform distribution). Compared to 2D GAN models, 3D-aware generative models need to build 3D object and camera models explicitly. So they have such requirements on the datasets.
> In our view, the reason why a 3D-aware generative model can work well on these datasets is that an implicit “canonical shape” (e.g., a forward-facing face for the FFHQ dataset) exists across the objects in different images, thus making the model easy to learn this “canonical shape” and “residual difference” on top of this canonical shape.
> For the natural landscape dataset used in StyleGAN, since the objects in images are not well aligned and the camera distribution is difficult to estimate, our model, as well as other 3D-aware models, would not work well. We will add this to the limitations and future work.
> One possible solution might be first training on synthetic data and then transferring it to diverse real images for future work.

---

### Public Comment · ~Miguel_A_Bautista1 · 2021-11-15
**Kudos and a related work reference seems to be missing**

I would like to congratulate the authors for putting together such a cool piece of work. Personally, I think that we are only starting to grasp the potential of 3D-aware generative models and this paper is a step in the right direction in my opinion. I would like to bring up to the authors that there seems to be a missing reference to relevant related work which uses a StyleGAN-like architecture in the generator and also "renders" feature maps that are then processed by a small convolutional block to produce the final RGB outputs.

@InProceedings{DeVries_2021_ICCV,
    author    = {DeVries, Terrance and Bautista, Miguel Angel and Srivastava, Nitish and Taylor, Graham W. and Susskind, Joshua M.},
    title     = {Unconstrained Scene Generation With Locally Conditioned Radiance Fields},
    booktitle = {Proceedings of the IEEE/CVF International Conference on Computer Vision (ICCV)},
    month     = {October},
    year      = {2021},
    pages     = {14304-14313}
}

---

> ### Author Response · Authors · 2021-11-21
> **Thanks for pointing out the missing reference**
>
> Thanks for pointing out the missing reference. We have cited GSN in the revision and discussed its differences from StyleNeRF.
> There are the following main differences.
>
> First, the objectives of these two works are different. StyleNeRF targets high-resolution (512^2 and above) and multi-view consistent synthesis of general objects like human faces, animal faces, and cars, while GSN is designed for image synthesis of indoor scenes. The results shown in GSN are in low resolution (64^2 or 128^2).
>
> Second, since the focuses of these two works are different, though the two methods share some similarities in the network components, the key issues tackled in these two works are different. For example, StyleNeRF studies the multi-view consistency issue while GSN does not. In addition, the idea of using a “2D floorplan” in GSN is tailored to indoor scenes where not much information needs to be encoded along the vertical axis to the “floor.” Therefore, this “2D floorplan” might be not suitable for a general object setting as addressed in StyleNeRF.
>
> Third, StyleNeRF is trained on RGB images only. In contrast, GSN needs RGB-D images for training, where the extra depth supervision is critical for its performance, as shown in their ablation studies.

---

### Author Response · Authors · 2021-11-21
**General response 1**

Thanks for the insightful feedback and suggestions. We now address the main concerns raised by the reviewers.

### 1 Novelty of this work (Reviewer TLAR, Reviewer gKfo)

We respectfully disagree with Reviewer TLAR and Reviewer gKfo that our method lacks novelty. If only looking at the network architecture itself, the main path of the proposed StyleNeRF does look similar to a concatenation of “pi-GAN”-like MLP with “GIRAFFE”-like CNN upsampler. Nevertheless, our model is not a simple combination of these two works. The design of StyleNeRF is derived from a different motivation: achieving high-resolution image synthesis and preserving multi-view consistency.

We start by building a generative NeRF model with only MLPs as building blocks to achieve this goal. Note that this model is unable to synthesize high-resolution images efficiently. Therefore, we step-by-step analyze and address the challenges in high-resolution image synthesis. We make two approximations in this generative NeRF model and reduce most of the computation by “early aggregation” and “up-sampling operations.” We also observe that our progressive training strategy is crucial for training this model robustly.

Furthermore, we propose several strategies such as “new upsampler” and “NeRF path regularization” to best preserve the multi-view consistency, which previous works have not studied.

Note that our model is different from GIRAFFE in the first place. GIRAFFE deploys a standard CNN renderer with a commonly used upsampler and 3x3 kernel, which is one of the main reasons GIRAFFE does not preserve multi-view consistency. In contrast, StyleNeRF is motivated and derived from the original NeRF model, and our 2D renderer after early aggregation is based on MLPs (1x1 convolution). With our carefully designed upsampler, we can finally achieve high-quality rendering results while preserving multi-view consistency.

---

### 2 Performance issues about baselines (Reviewer TLAR, Reviewer gKfo)

First of all, for a fair comparison, we used either the pretrained checkpoints (if available) or the officially released codes to run the same datasets with their recommended settings and hyper-parameters. We also tried multiple sets of parameters for the results where the default setting does not work well (e.g., Pi-GAN’s result of the Compcars dataset).
We think there are two reasons why the results of Pi-GAN and GIRAFFE showed in our paper look worse than those in their original papers:

- **Different Datasets:** As the goal of our paper focuses on high-resolution images, we mainly chose the datasets of 512^2 or 1024^2 images in our experiments, i.e., FFHQ, AFHQ, and MetFaces, which are different from those used in other papers. These datasets are more challenging (e.g.objects in images are diverse), thus making pi-GAN or other baselines fail to produce good results.

- **Different Visualization:** When compared with GIRAFFE on FFHQ and CompCars, the quantitative results in terms of FID are indeed similar to those shown in their original paper. The drop in the visual results lies in how the results were visualized. The GIRAFFE paper only showed their results from sparse viewpoints, while we showed their results from denser viewpoints where we observed more inconsistent artifacts in their results.

---

> ### Author Response · Authors · 2021-11-21
> **General response 2**
>
> ### 1 Additional quantitative results on multi-view consistency (Reviewer  gKfo)
> In our submission, we mainly focused on the qualitative comparison for the multi-view consistency of each approach (Fig 4 for comparison and Appendix Fig 12 for COLMAP reconstruction).
> It is a good idea to measure the consistency quantitatively. However, the metric used in GRAF for measuring multi-view consistency needs 3D ground-truth geometry for calculation, so it can only be applied to synthetic data, not real data where the ground-truth geometry is not available.
>
> Instead, here we propose a new method for measuring multi-view consistency without the requirement of 3D shapes. First, we randomly sample 1000 seeds, and for each seed, render a sequence of 9 images following the same trajectory. Then we take four images as conditioning and predict pseudo ground truth images given the remaining five camera poses using a pre-trained IBRNet (Wang et al., 2021), a recently proposed image-based rendering method for generic view interpolation function generalizes to novel scenes. Then we measure the average image reconstruction scores (PSNR, SSIM and LPIPS) between the pseudo ground truth and remaining five images generated by a 3D-aware generative model (i.e., StyleNeRF and baselines). Since the IBRNet model is fixed for all methods, the reconstruction scores can be used as a metric to measure the multi-view consistency of different methods.
>
> We tested on the FFHQ 256x256 and AFHQ 256x256 dataset and showed the results as follows:
>
> | Dataset | Metrics | GIRAFFE | pi-GAN | StyleNeRF |
> |:---------|:----------:|:--------------:|:----------:|:--------------:|
> |             | PSNR | 25.8 | 29.5 | 29.0 |
> |  FFHQ | SSIM | 0.81 | 0.92 | 0.89 |
> |             | LPIPS | 0.13 | 0.04 | 0.08 |
> |             | PSNR | 26.2 | 28.5 | 26.8 |
> |  AFHQ | SSIM | 0.75 | 0.86 | 0.80 |
> |             | LPIPS | 0.25 | 0.12 | 0.13 |
>
>
> For the results,
> the proposed StyleNeRF
> achieves much better consistency than GIRAFFE, while it is comparable to pi-GAN, the state-of-the-art pure
> NeRF-based model, in terms of multi-view consistency. Meanwhile, StyleNeRF can generate high-resolution images efficiently, which cannot be achieved by existing pure NeRF-based models.
> We have also included this comparison in Appendix D in the revised paper.
> ---
> ### 2 Dataset bias (gaze following effect) (Reviewer Qm5b and Reviewer MWb6)
> Both Reviewer Qm5b and Reviewer MWb6 mentioned that in the results of FFHQ, the rendered eyes are always following the camera direction.
>
> First of all, this is due to the dataset bias. The dataset used for training is **NOT** an actual multi-view image dataset, but a collection of images where each person is only shown once from a specific viewpoint. The characters’ eyes are almost always looking at the camera. A generative model can easily capture such a spurious correlation between the eye-looking direction and camera position. Hence, such phenomena exist not only in our model but also in other baseline methods.
>
> ---
> ### 3 Notations and typos:
> We apologize for the inconsistent and unexplained notations and typos in the paper. In Fig 3, K_fg and K_bg are the numbers of sample points along the ray for foreground and background, respectively. K_fg and K_bg are not fixed across different datasets. Besides, we adopt a similar setting as in pi-GAN. We have added these explanation and fixed the typos in the revision.
>
> ----
> - Reference: Wang, Qianqian, et al. “Ibrnet: Learning multi-view image-based rendering.” Proceedings of the IEEE/CVF Conference on Computer Vision and Pattern Recognition. 2021.

---

### Decision · Program_Chairs · 2022-01-20

**Decision:**

Accept (Poster)

**Comment:**

An interesting paper on combining NerFs with StyleGAN to get high-quality and high-resolution 3d aware generative models. The results are very good visually and also allow interactive speeds.  The technique is natural and concurrent papers are proposing variations

The reviewers identified a few limitations including that the nerf does not have a viewing direction and also seems limited to aligned objects with a common structure, like faces. Still the results are very interesting and suitable for publication.